# Integrated hydrogeological and hydrogeochemical dataset of an alpine catchment in the northern Qinghai-Tibet Plateau

Zhao Pan[1], Rui Ma[1*], Ziyong Sun[1], Yalu Hu[1], Qixin Chang[2], Mengyan Ge[1], Shuo Wang[1], Jianwei Bu[1], Xiang Long[1], Yanxi Pan[1], and Lusong Zhao[1]

1 Hubei Key Laboratory of Yangtze River Basin Environmental Aquatic Science, School of Environmental Studies, and State Key Laboratory of Biogeology and Environmental Geology, China University of Geosciences, Wuhan 430078, China
2 College of Environment and Civil Engineering, Chengdu University of Technology, Chengdu 610059, China

*Correspondence to*: Rui Ma (rma@cug.edu.cn)

**Abstract.** Climate warming has significantly changed the hydrological cycle in cold regions, especially in areas with permafrost or seasonal frost. Groundwater flow and its interactions with surface water are essential components of the hydrological process. However, few studies or modeling works have been based on long-term field observations of groundwater level, temperature, hydrogeochemistry or isotopic tracers from boreholes due to obstacles such as remote locations, limited infrastructure, and harsh work conditions. In the Hulugou catchment, an alpine catchment in the headwater region of the Heihe River on the northern Qinghai-Tibet Plateau (QTP), we drilled four sets of depth-specific wells and monitored the groundwater levels and temperatures at different depths. Surface water (including river water, glacier meltwater, and snow meltwater), precipitation, groundwater from boreholes, spring water, and soil water were sampled to measure the abundances of major and minor elements, dissolved organic carbon (DOC), and stable and radioactive isotopes at 64 sites. This study provides a dataset of these groundwater parameters spanning six consecutive years of monitoring/measurements. These data can be used to investigate groundwater flow processes and groundwater-surface water interactions on the QTP under global climate change. The dataset provided in this paper can be obtained at

## 1 Introduction

The Qinghai-Tibet Plateau (QTP) is known as the "Water Tower of Asia" because it comprises the headwater regions of many large rivers in Asia, including the Yangtze, Yellow, Lancang-Mekong, Nujiang-Salween, Yarlung Tsangpo, Ganges, Indus, Tarim and other rivers (Qiu, 2008; Immerzeel et al.,
2010). At present, the total amount of water storage on the QTP (including glacier reserves, lake water, and runoff from the outlets of major rivers) has been initially estimated to exceed nine trillion cubic meters (Li et al., 2019; Pritchard, 2019; Liu et al., 2020). However, under the background of global warming, not only has the temperature of the QTP gradually increased over the past 50 years (Hartmann et al., 2013; Yao et al., 2013), but the warming rate has also been slowly increasing (Chen et al., 2015; Kuang and
Jiao, 2016). Therefore, the QTP has been experiencing a series of environmental changes, such as permafrost degradation, continuously decreasing snow cover, and glacier and lake shrinkage (Jin et al., 2011; Yao et al., 2012; Liu et al., 2015; Huang et al., 2017; Xu et al., 2017; Bibi et al., 2018; Ran et al., 2018; Yao et al., 2019b; Bian et al., 2020). These changes have vastly impacted the hydrological system on the QTP, affected the living environments of 1.7 billion people, and caused economic losses of up to
$12.7 trillion (Yao et al., 2019a).

Previous studies have shown that groundwater plays important roles in maintaining runoff, not only in low flow periods but also in high flow periods, of rivers in permafrost regions and regulating their flow regimes (Walvoord et al., 2012; Carey et al., 2013; Ma et al., 2017; Evans et al., 2020; Ma et al., 2021a; Wang et al., 2021). Some studies have suggested that increased groundwater temperatures caused by
climate warming adversely affect the biogeochemical process in aquifers, resulting in a decline in groundwater quality and thus affecting the use of water resources (Green et al., 2011; O'Donnell et al., 2012; Sharma et al., 2012; Cochand et al., 2019). In the case of global warming, interactions between groundwater and surface water may cause the release of carbon trapped in permafrost and aggravate the

greenhouse effect (Harlan, 1973; Solomon et al., 2007; Schaefer et al., 2011; Wisser et al., 2011; Mckenzie and Voss, 2013; Connolly et al., 2020; Behnke et al., 2021). It was also reported that saturated sediments that are not frozen year-round due to the heat released by groundwater movement in the subsurface layer and the thermal insulation effect of surface ice layers provide interstitial habitats for groundwater fauna (Schohl and Ettema, 1990; Clark and Lauriol, 1997; Alekseyev, 2015; Huryn et al., 2020; Terry et al., 2020). All these findings undoubtedly confirm the importance of groundwater flow in hydrological systems cold regions.

However, existing studies on groundwater and surface water interactions in cold regions have mainly focused on the Arctic which is generally characterized by the presence of continuous permafrost and the dominance of spring snowmelt in groundwater recharge (Mcclymont et al., 2010). In contrast to the Arctic, the catchments on the QTP are generally characterized by complex combinations of continuous permafrost, discontinuous permafrost, island permafrost and seasonally frozen ground due to the tremendous topographical and elevational differences in the same catchment (Cheng and Jin, 2013; Chang et al., 2018). Furthermore, the Asian monsoon climate causes most of the hydrological inputs on the QTP to occur in summer and autumn when precipitation and glacier meltwater are the largest (Lu et al., 2004; Chang et al., 2018). Different combinations of permafrost, seasonally frozen ground and hydroclimatic conditions lead to more complex interactions between groundwater and surface water on the QTP (Woo, 2012), which need to be studied in greater detail to predict the response of hydrological processes to climate change in the future. Current studies on the interactions between groundwater and surface water on the QTP have focused on tracing flow paths using different types of isotopic and geochemical data or building numerical water-heat coupling models to explore the influence of climate change on hydrological processes (Ma et al., 2009; Ge et al., 2011; Evans et al., 2015; Hu et al., 2019; Li et al., 2020a; Yang and Wang, 2020; Tan et al., 2021). Although significant progress has been made in studying hydrological processes on the QTP, including changes of streamflow compositions or surface flow regime under the impact of climate change and frozen soil degradation (Wang et al., 2018; Zheng et al., 2018; Cuo et al.,

2019; Xu et al., 2019; Shen et al., 2020), little is known about the groundwater system or the processes that control groundwater and surface water interactions due to challenges from the poor field conditions and weak infrastructure on the QTP (Yao et al., 2019a). Most existing studies have used spring water and baseflow measurements in winter to represent groundwater and have rarely directly observed groundwater indicators through boreholes (Pu et al., 2017; Gui et al., 2019; Li et al., 2020b; Pu et al., 2021). The coupled groundwater flow and heat transport models employed in previous studies to represent permafrost regions on the QTP have mainly focused on scenarios in which the conceptual models are tested; however, the numerical models are normally lack of verification with actual field data (Ge et al., 2011; Evans et al., 2015). Existing large-scale groundwater flow models on the QTP did not incorporate the effect of freeze-thaw processes induced by heat transport processes on the hydrological cycle due to the model complexity (Yao et al., 2017; Yao et al., 2021). To the beginning of this study, no research site has been established on the QTP that focuses on groundwater flow or its interaction with surface water, with systematic monitoring of physicochemical indicators of groundwater and surface water and freeze-thaw processes of frozen soil. To fill this gap, we established a systematic monitoring site in the upper reaches of the Heihe River on the northern QTP.

Specifically, this paper introduces a monitoring system for the groundwater level, ground temperature, water chemistry, and isotopic compositions of various water bodies in an alpine catchment. In Section 2, the general setting of the study area is introduced in detail. In Section 3, the layout of the monitoring wells, the lithology of the boreholes, and the mineralogical compositions of the cores are described. In addition, the dynamics of groundwater level and ground temperature are also shown. In Section 4, the methods used for the collection, preservation, and analysis of samples representing various water bodies are described. The general characteristics of water chemistry and isotopes in different waters are shown in Section 5. Finally, the methods for obtaining raw data mentioned in this article are provided in Section 6, and future research prospects and a summary of the whole article are provided in Section 7.

## 2 Study area

Our study area, the Hulugou catchment, is one of the typical catchments located in the headwaters of the Heihe River in the northern QTP (Fig. 1). The catchment occupies an area of 23.1 km$^2$ (99°50′37″–99°53′54″E and 38°12′14″–38°16′23″N). The elevation ranges from 2960 to 4800 m a.s.l. The study area has a continental climate, with warm, humid summers and cold, dry winters. The mean annual temperature is -3.1 °C, and the mean annual precipitation is 403.4 mm, approximately 70% of which is concentrated from July to September (Chen et al., 2014a; Chen et al., 2014b). The water surface evaporation was 1231 mm in 2013 (Yang et al., 2013). The discharge from the catchment was approximately 567.7 mm/year in 2012 (Chen et al., 2014a; Chen et al., 2014b). The hydrometeorological monitoring network of the catchment is composed of five automatic meteorological stations and one stream gauging station (Fig. 1), which are maintained and operated by the Qilian Alpine Ecology and Hydrology Research Station, Northwest Institute of Eco-Environment and Resources, Chinese Academy of Sciences (http://hhsy.casnw.net). Researchers with a reasonable need for the precipitation, air temperature, and runoff data for the study area may request them by email. The website for the specific contact information is http://hhsy.casnw.net/lxwm/index.shtml.

Two main tributaries originate from the southern glaciers in the study area (Fig. 1). These tributaries are mainly fed by glacier and snow meltwater, ice lakes, and springs in the high mountains (Yang et al., 2013; Chang et al., 2018; Hu et al., 2019). In the warm season of each year (May to September), these tributaries are sustained; they are intermittently dry up during the rest of the year (Yang et al., 2013; Chang et al., 2018; Hu et al., 2019). The two tributaries converge into a single channel at the northern end of a piedmont alluvial plain and finally flow into the main channel of the Heihe River. The groundwater and surface water frequently interact in the study area (Ma et al., 2017; Chang et al., 2018; Hu et al., 2019).

The landforms above 4200 m a.s.l. in the study area are mainly shaped by mountain glaciers. The area comprising modern glaciers and permanent snow cover is approximately 1.62 km$^2$ (Ma et al., 2017;

Chang et al., 2018; Hu et al., 2019) (Fig. 2a). Many moraine sediments are distributed on the surface in this area, constituting a porous aquifer of moraine breccia with large pores and good hydraulic connectivity (Chang et al., 2018) (Fig. 2b). According to field investigations, the aquifer exhibits high permeability and provides a good channel for groundwater flow (Ma et al., 2017; Chang et al., 2018). In the warm season, the aquifer is recharged by glacier and snow meltwater and rainfall; this water is rapidly discharged to nearby rivers and provides lateral recharge for low-elevation aquifers (Ma et al., 2017).

In the elevation range of 3500 to 4200 m a.s.l., permafrost is distributed discontinuously, and the active layer is ~2 m thick (Ma et al., 2017). The drilling data show that the permafrost is approximately 20 m thick (Ma et al., 2017; Hu et al., 2019). There is a porous aquifer consisting of argillaceous gravel on the planation surface at the top of the hill in the permafrost area (Ma et al., 2021a) (Fig. 2c). In the warm season, the groundwater in this aquifer is mainly recharged by the infiltration of glacier and snow meltwater, precipitation, and groundwater from the higher-elevation aquifer. It exists as suprapermafrost, intrapermafrost, and subpermafrost groundwaters (Ma et al., 2021a)and typically discharges to the tributaries as springs at the foot of the slope.

Seasonal frost is mainly distributed in areas below 3500 m a.s.l. The maximum freezing depth is approximately 3 m; freezing to this depth occurs in January (Ma et al., 2017). Alluvial sediments in the piedmont plain constitute a porous aquifer containing sands and gravels (Fig. 2d). The thickness of this aquifer is 20–50 m, and it can store water in summer and maintain baseflow in winter to regulate streamflows (Ma et al., 2021a). In this area, the stream and groundwater are hydraulically well connected. Groundwater in the aquifer is recharged by the infiltration from the stream and the lateral groundwater runoff from the adjacent mountains, and finally discharges into the stream again at lower elevation in the valley or through springs near the catchment outlet (Ma et al., 2017; Chang et al., 2018). The groundwater mainly flows from south to north, consistent with the relief.

The permafrost zone (3500–4200 m a.s.l.) is dominated by alpine meadow, and the vegetation coverage is ~80 % (Chen et al., 2014b; Yang et al., 2015). The seasonal frost zone (below 3500 m a.s.l.)

is dominated by alpine meadow and alpine shrubs, and the vegetation coverage is ~95 % (Liu et al., 2012; Chen et al., 2014b; Yang et al., 2015).

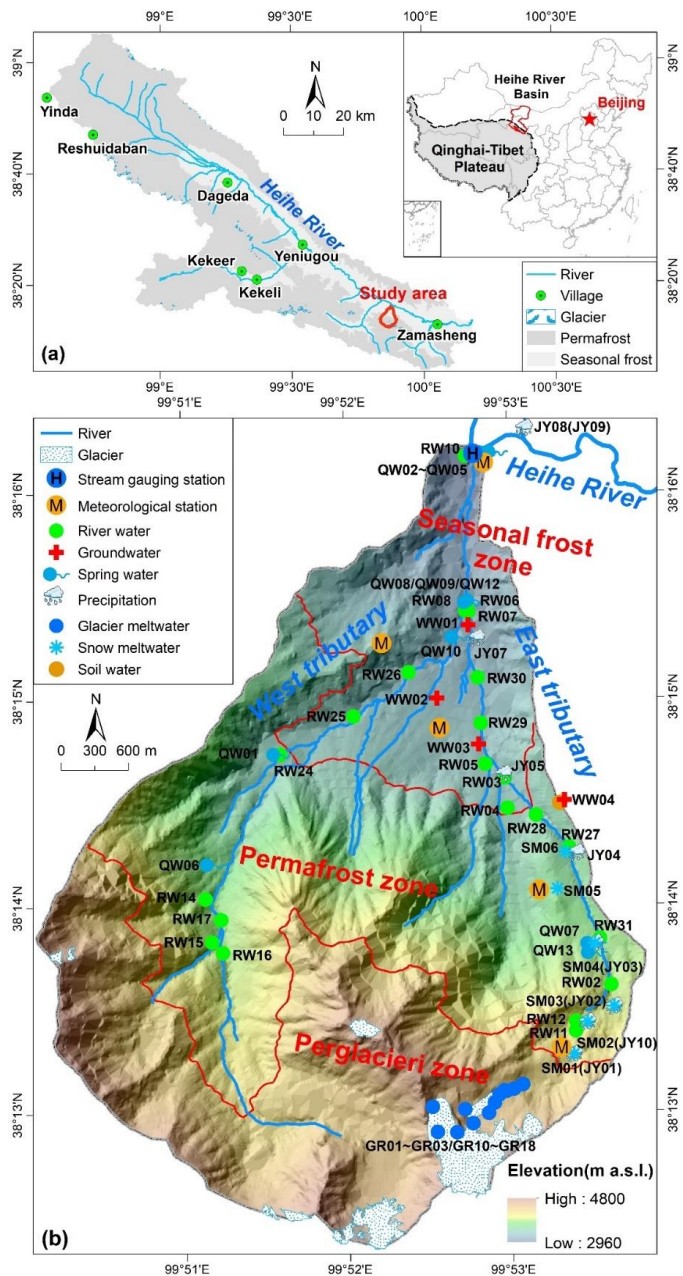

Figure 1. (a) The location of the study area in the headwater regions of the Heihe River; and (b) the Hulugou catchment showing the topography and the monitoring and sampling sites. The

**permafrost distribution and elevation data were obtained from the National Tibetan Plateau Data Center (http://data.tpdc.ac.cn), and the resolution of elevation data is 5 meters.**

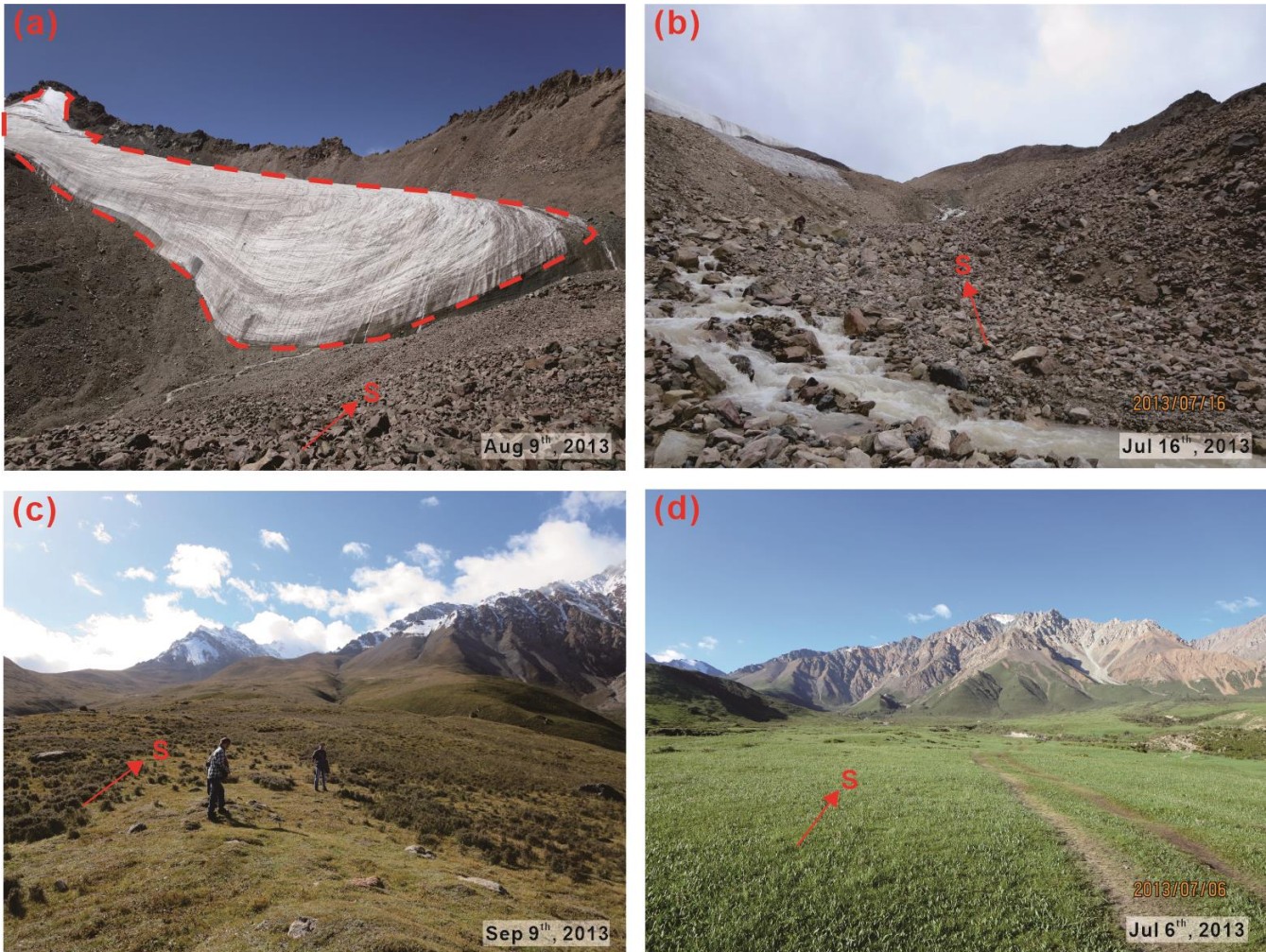

**Figure 2. Pictures showing (a) the glaciers in the south part of the study area, (b) the moraine sediments in the periglacial zone, (c) the planation surface in the permafrost zone, and (d) the piedmont alluvial plain in the seasonal frost zone. The letter "S" denotes the south direction.**

**3 The monitoring system**

The general principle behind the design of the field monitoring system is that the system can be used to obtain different kinds of hydrological and hydrogeochemical data for various hydrological components

within the catchment such as glacier-snow meltwater, groundwater, and stream water during different periods of the freeze-thaw cycle and at different locations along primary flow paths. With these data, it is possible to explore the hydrological processes and associated biogeochemical processes under the influence of the freeze-thaw cycle at the catchment scale.

## 3.1 Layout of well groups

Four well groups were constructed in the study area, of which well groups WW01, WW02 and WW03 are located in the seasonal frost and well group WW04 is located in the permafrost area (Fig. 1). The specific locations of these well groups are as follows: WW03 is at the top of the piedmont plain near the southern bedrock mountains, where is the recharge zone of the groundwater in the seasonally frozen area; WW02, located in the middle of the piedmont plain, was supposed to capture the groundwater flow in the flow-through zone, but the water table is lower than well screen depths thus the groundwater level was not monitored; WW01 is near the intersection of the east and west tributaries in the northern part of the study area where is the groundwater discharge zone; WW04 is located on a planation-caused terrace in the eastern mountains. The details of these well groups are listed in Table 1, including their longitudinal and latitudinal coordinates, elevation, number of boreholes and depth of each borehole, and the depths of the pressure and temperature sensors deployed. The field scenes for the layout of each well group are shown in Fig. 3. Each well group includes five or seven depth-specific wells to monitor the groundwater level and temperature at different depths. The screen depth for each well is given in Table 1. High-density polyethylene (HDPE) pipe was used for the well walls, which is expected to have little effect on the measured groundwater properties. For each well group, the well with a depth of 3 m was used to monitor soil temperature. For the remaining wells, the bottom 0.5 m part was screened with a filter pipe.

**Table 1. The number, coordinates, elevation, borehole depth, and monitoring information of each well group in the study area (the "√" symbol indicates that a sensor was placed at the bottom of the borehole, and the "×" symbol indicates that no sensor was placed).**

| Well group no. | WW01 | | | | |
|---|---|---|---|---|---|
| Coordinates and elevation | N: 38°15′21.27″, E: 99°52′45.38″, 3144 m a.s.l. | | | | |
| Borehole no. | WW01–01 | WW01–02 | WW01–03 | WW01–04 | WW01–05 |
| Borehole depth (m) | 25 | 15 | 10 | 5 | 3 |
| Pressure sensor | √ | √ | √ | √ | × |
| Depth of temperature sensor (m) | 23 | 15 | 10 | 5 | 0.2, 0.5, 1, 1.5, 2, 3 |
| Well group no. | WW02 | | | | |
| Coordinates and elevation | N: 38°15′0.03″, E: 99°52′33.68″, 3250 m a.s.l. | | | | |
| Borehole no. | WW02–01 | WW02–02 | WW02–03 | WW02–04 | WW02–05 |
| Borehole depth (m) | 30 | 15 | 10 | 5 | 3 |
| Pressure sensor | × | × | × | × | × |
| Depth of temperature sensor (m) | 30 | 15 | 10 | 5 | 0.2, 0.5, 1, 1.5, 2, 3 |
| Well group no. | WW03 | | | | |
| Coordinates and elevation | N: 38°14′46.57″, E: 99°52′48.87″, 3297 m a.s.l. | | | | |
| Borehole no. | WW03–01 | WW03–02 | WW03–03 | WW03–04 | WW03–05 |
| Borehole depth (m) | 30 | 20 | 10 | 5 | 3 |
| Pressure sensor | √ | √ | × | × | × |
| Depth of temperature sensor (m) | 29 | 18.5 | 10 | 5 | 0.2, 0.5, 1, 1.5, 2, 3 |

| Well group no. | WW04 | | | | | | |
|---|---|---|---|---|---|---|---|
| Coordinates and elevation | N: 38°14′30.25″, E: 99°53′20.21″, 3501 m a.s.l. | | | | | | |
| Borehole no. | WW04–01 | WW04–02 | WW04–03 | WW04–04 | WW04–05 | WW04-06 | WW04-07 |
| Borehole depth (m) | 24.3 | 17.5 | 12 | 7 | 5 | 3 | 1.5 |
| Pressure sensor | √ | × | × | × | × | × | √ |
| Depth of temperature sensor (m) | 23.6 | 17.2 | 11.8 | 6.7 | 4.7 | 0.2, 0.5, 1, 1.5, 2, 3 | 1.5 |

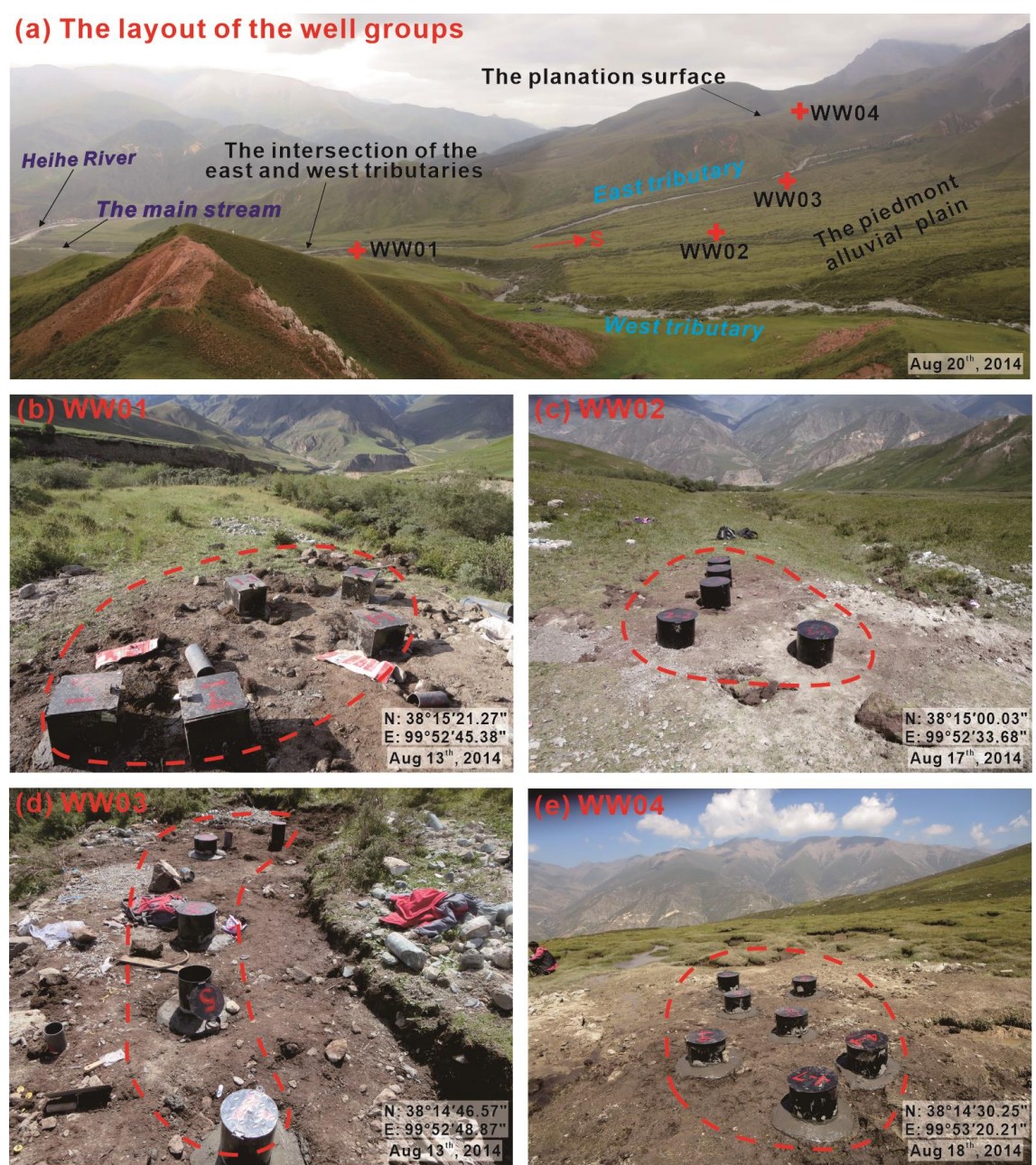

**Figure 3. Pictures showing the field scenes for (a) the location and layout of the four well groups in the study area, (b) the layout of wells in the well group WW01, (c) the layout of wells in the well group WW02, (d) the layout of wells in the well group WW03, and (e) the layout of wells in the well group WW04. The letter "S" denotes the south direction.**

## 3.2 Sediments and their mineral composition of the borehole cores

Core sampling and lithology identification were conducted at different depths during the drilling processes (Fig. 4), and the results are shown in Fig. 5. In the seasonal frost area of the piedmont alluvial plain, the sediments are mainly composed of mud-bearing pebble gravels that are very loose and poorly sorted (Ma et al., 2017; Chang et al., 2018). With the same drilling depth, the well groups WW01 and WW02 were not drilled to bedrock, while the well group WW03 was drilled to weathered sandstone bedrock, indicating that the thickness of the aquifer located at the top of the piedmont alluvial plain is thinner than that in the middle and lower parts (Fig. 5). In addition, a clay layer with a thickness of 3–6 m was found in all the well groups WW01, WW02 and WW03 (Fig. 5).

As revealed by the boreholes in the well group WW04 on the planation surface of the permafrost region, the perennially frozen layer is distributed at depths between 2 and 20 m underground (Fig. 5) and mainly consists of sand, gravel, and ice. The active layer is ~2 m thick and is composed of silt clay with small gravel (Fig. 5). Below the depth of 20–24.3 m is the unfrozen subpermafrost aquifer, consisting mainly of sand and gravel (Fig. 5). An unfrozen intrapermafrost aquifer was also revealed, located at a depth of 12 m underground and consisting of a clay layer with a thickness of approximately 0.2 m (Fig. 5).

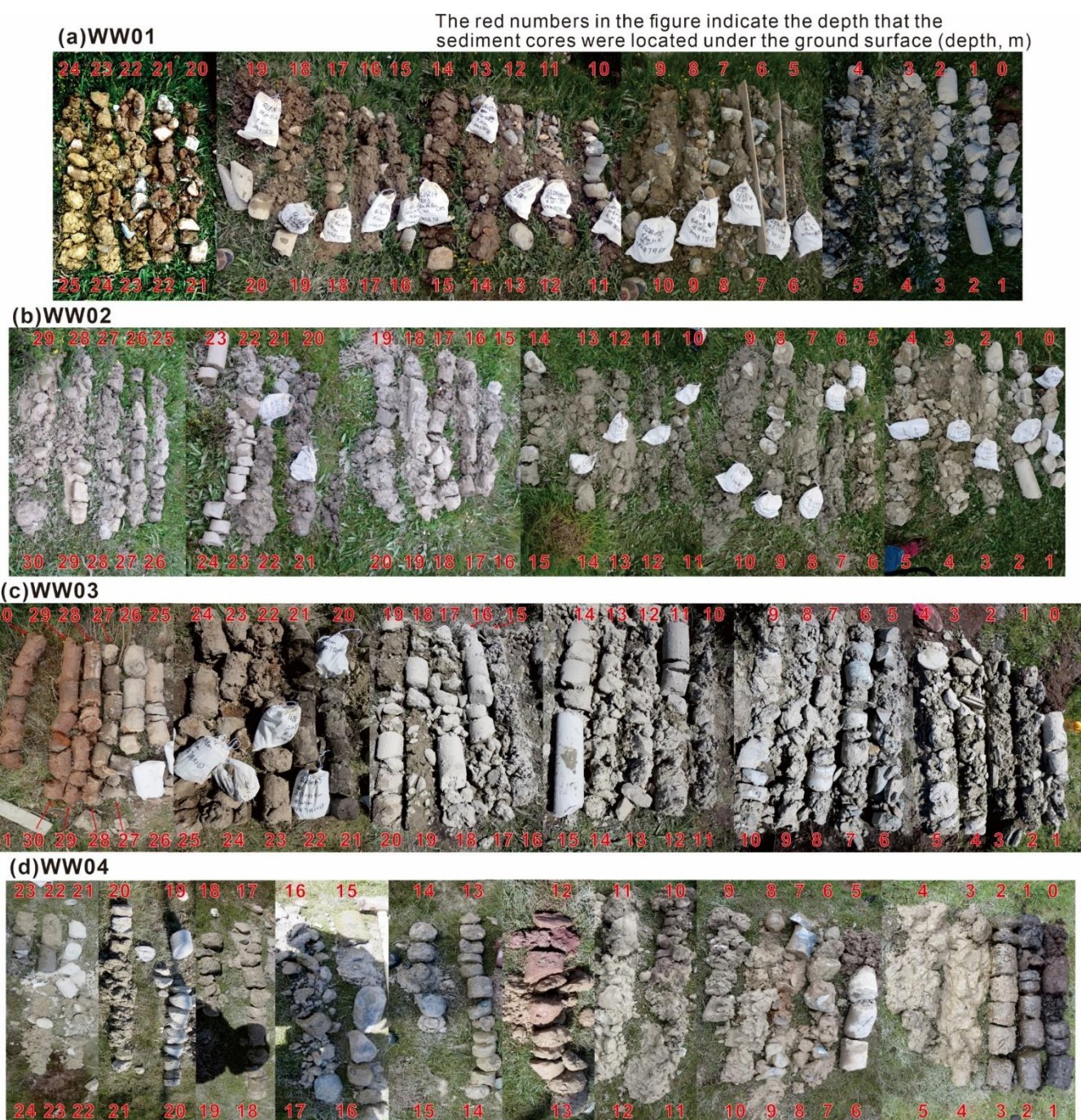

The red numbers in the figure indicate the depth that the sediment cores were located under the ground surface (depth, m)

**Figure 4. Pictures showing the lithology of cores at different depths belowground from the well groups (a) WW01, (b) WW02, (c)WW03, and (d) WW04.**

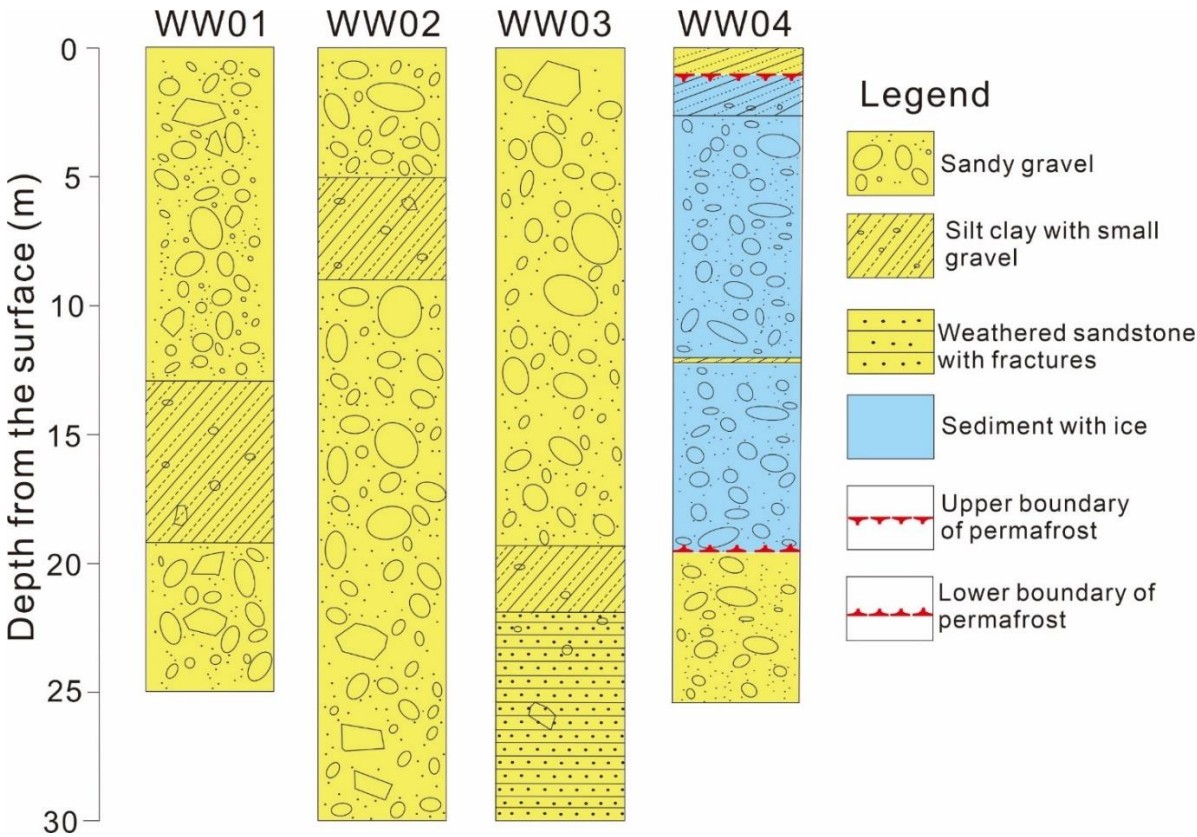

**Figure 5. Lithology diagram of the cores from the four well groups (Modified from Ma et al., 2017).**

During the drilling of the well groups WW03 and WW04, core samples were collected at 5-m intervals or at depths where the lithology changed. The length of each core sample was approximately 10 cm. The core samples were sealed and placed in shading bags and then stored at -20 °C. Then, the mineral components and their contents of each sample were analyzed by X-ray diffraction (X'Pert PRO) in the State Key Laboratory of Geological Processes and Mineral Resources at the China University of Geosciences.

The results of the mineral components and their proportion of the core samples are listed in Table 2. Chlorite, illite, quartz and K-feldspar are the dominant minerals in these samples, accounting for more than 80% of all minerals. Calcite and dolomite account for 0% to 10% and 2% to 25% of the total mineral contents, respectively, while the proportion of amphibole does not exceed 8%.

**Table 2. Mineral components and their proportions of the core samples.**

| Borehole no. | Sample depth (m) | Percentage of mineral components (%) | | | | | | |
|---|---|---|---|---|---|---|---|---|
| | | Chlorite | Illite | Quartz | K-feldspar | Calcite | Dolomite | Amphibole |
| WW03–01 | 0–0.2 | 25 | 20 | 30 | 7 | 10 | 8 | 0 |
| | 2.6–2.8 | 25 | 15 | 30 | 10 | 10 | 10 | 0 |
| | 7.5–7.7 | 25 | 15 | 25 | 5 | 25 | 5 | 0 |
| | 12.6–12.8 | 25 | 15 | 33 | 10 | 10 | 5 | 2 |
| | 17.4–17.6 | 25 | 15 | 30 | 18 | 10 | 2 | 0 |
| | 21.8–22.0 | 25 | 15 | 40 | 18 | 2 | 0 | 0 |
| | 24.6–24.8 | 25 | 15 | 37 | 10 | 8 | 5 | 0 |
| | 29.6–29.8 | 25 | 15 | 40 | 5 | 15 | 0 | 0 |
| WW04–01 | 0.2–0.4 | 25 | 15 | 45 | 12 | 3 | 0 | 0 |
| | 1.6–2.0 | 25 | 15 | 43 | 15 | 0 | 0 | 2 |
| | 2.0–2.3 | 25 | 15 | 45 | 15 | 0 | 0 | 0 |
| | 5.0–5.2 | 25 | 15 | 29 | 15 | 8 | 8 | 0 |
| | 10.4–10.6 | 15 | 25 | 30 | 15 | 5 | 10 | 0 |
| | 15.6–15.8 | 25 | 15 | 25 | 15 | 8 | 10 | 2 |
| | 21.1–21.3 | 25 | 15 | 25 | 22 | 5 | 0 | 8 |
| | 24.2–24.4 | 25 | 15 | 28 | 15 | 10 | 5 | 2 |

## 3.3 Monitoring of groundwater levels and ground temperatures

During the installation of the monitoring systems, pressure sensors (HOBO U20–001–02, ONSET, USA) were placed at the screening locations of wells with depths of 25 m, 15 m, and 10 m in well group WW01, depths of 30 m and 20 m in well group WW03, and depths of 24.3 m and 1.5 m in well group WW04 to monitor the groundwater level at 30-minute intervals (Table 1). No groundwater was observed

in well group WW02. The operation range of the pressure sensors is approximately 0 to 33.6 m of water depth at 3000 m a.s.l. These sensors have a measurement accuracy of ± 1.5 cm, and a resolution of 0.41 cm. Temperature sensors (HOBO S–TMB–M0017, ONSET, USA) were also installed at different depths to monitor the ground temperature at 30-minute intervals. The details of these sensors are summarized in Table 1. Extension cables (HOBO S–EXT–M025, ONSET, USA) were used to measure temperatures at depths greater than 17 m. The measurement range of the temperature sensors is from -40 °C to 100 °C. The measurement accuracy is ± 0.2 °C with a resolution of ± 0.03 °C for temperatures from 0 to 50 °C. All pressure and temperature sensors were calibrated in the laboratory before use (Text S1). The monitoring of groundwater level and ground temperature began in September 2014 and lasted through September 2020. The periods with available data are shown in Table 3 for the groundwater level and Table 4 for ground temperatures.

**Table 3. The periods with available groundwater level data.**

| Well group no. | Borehole depth (m) | Periods with available data |
|---|---|---|
| WW01 | 5 | 2014/08–2014/12, 2015/07–2015/12, 2016/07–2016/12, 2018/08–2018/12, 2019/07–2019/12 |
| | 10 | 2014/08–2015/01, 2015/06–2016/01, 2016/06–2017/01, 2017/06–2018/01, 2018/06–2019/01, 2019/06–2020/01, 2020/06–2020/08 |
| | 15 | 2014/08–2015/02, 2015/06–2016/02, 2016/06–2017/02, 2017/06–2018/02, 2018/06–2019/01, 2019/06–2019/09 |
| | 25 | 2014/08–2015/02, 2015/06–2015/07, 2015/09–2019/08 |
| WW03 | 20 | 2014/08–2014/10, 2015/04–2015/06, 2015/07–2020/08 |
| | 30 | 2014/08–2014/12, 2015/04–2020/08 |
| WW04 | 1.5 | 2014/09–2015/01, 2015/07–2016/01, 2016/09–2017/01, 2017/04–2017/07, 2018/07–2019/12, 2019/08–2019/12, 2020/07–2020/08 |
| | 24.3 | 2015/08–2016/07 |

**Table 4. The periods with available ground temperature data.**

| Well group no. | Depths (m) | Periods with available data |
|---|---|---|
| WW01 | 0.2, 1 | 2014/09–2017/07, 2018/07–2020/07 |
| | 0.5, 1.5 | 2014/09–2017/07, 2018/07–2019/09 |
| | 2, 3, 5 | 2014/09–2018/11 |
| | 10, 13, 23 | 2014/09–2019/07 |
| WW02 | 0.2, 1, 1.5 | 2014/09–2019/05, 2019/07–2020/08 |
| | 0.5 | 2014/09–2018/09, 2019/07–2020/08 |
| | 2, 3, 5 | 2014/09–2014/10, 2015/01–2017/04, 2017/08–2020/08 |
| | 10, 15, 30 | 2014/09–2020/08 |
| WW03 | 0.2, 0.5, 1, 1.5 | 2014/09–2016/07, 2016/9–2020/06, 2020/07–2020/08 |
| | 2, 3, 5, 10, 18.5, 29 | 2014/09–2020/08 |
| WW04 | 0.2, 0.5, 1, 1.5 | 2014/09–2020/08 |
| | 2, 3, 4.7, 6.7 | 2014/09–2019/04, 2019/07–2020/06 |
| | 11.8, 17.2 | 2014/09–2019/07 |
| | 23.6 | 2014/09–2015/09 |

The dynamics changes of groundwater level measured in each well from July 2014 to September 2020 are shown in Fig. 6. From June to September, the groundwater level in WW03 fluctuated widely while that in WW01 was relatively stable. The groundwater level in both WW03 and WW01 was much lower from November to May than from June to September within each year. The groundwater level in the 1.5-m deep well in well group WW04, which was located in the permafrost region, was close to the surface from June to October and dropped rapidly from the end of October to November, while that in the 24.3-m deep well varied between 20.3 m and 23.4 m.

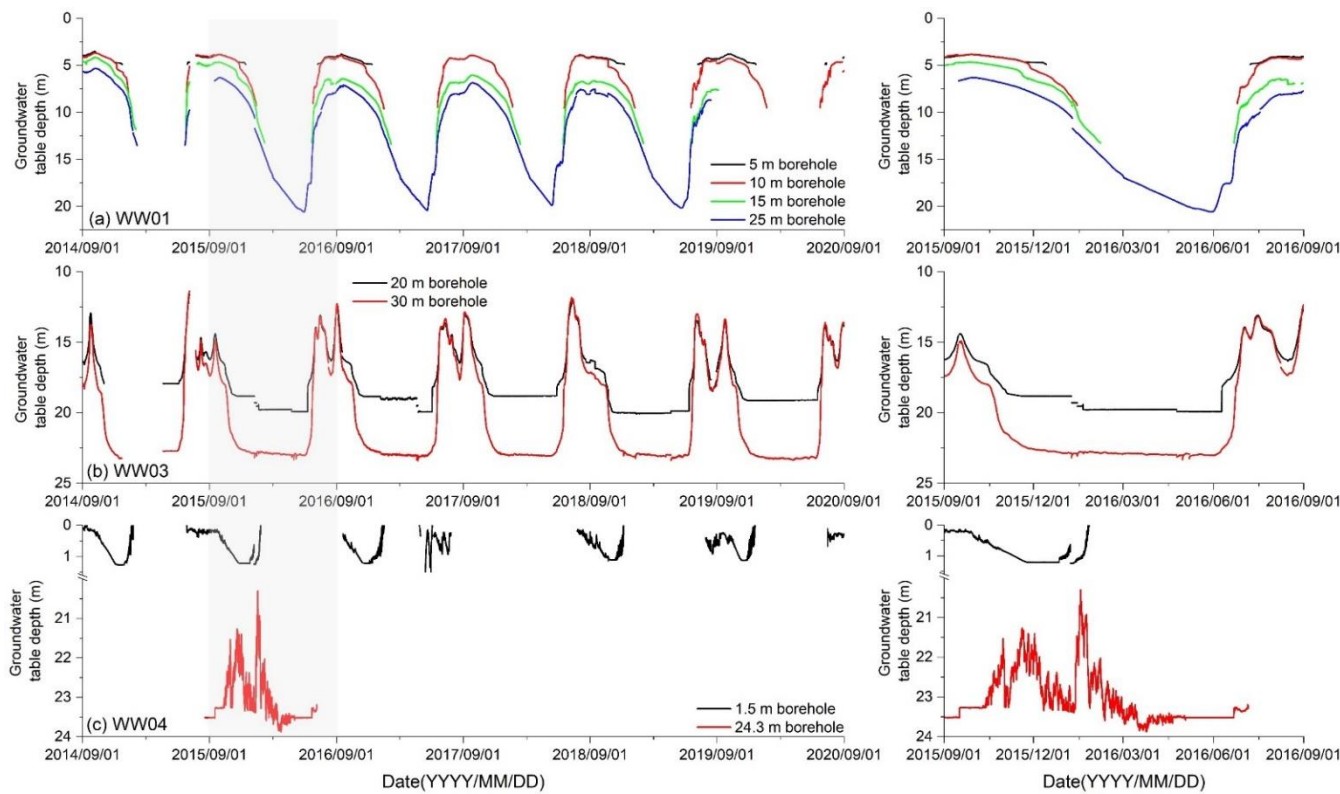

**Figure 6. The left panels show the dynamics of groundwater depth over six consecutive years, while the right panels highlight the variation in groundwater depth within a hydrological year.**

Fig. 7 shows the dynamics of ground temperature with time. The ground temperature fluctuated wildly at near-surface depths. According to the ground temperature measured at different depths in well groups WW01, WW02 and WW03, the maximum freezing depth of seasonal frost ranged from 3 to 5 m, and and the intra-annual fluctuations in ground temperature basically disappeared at depths greater than 10 m. In contrast, the ground temperature measured in well group WW04 fluctuated mainly at depths above 2 m, while it remained near 0 °C at deeper depths.

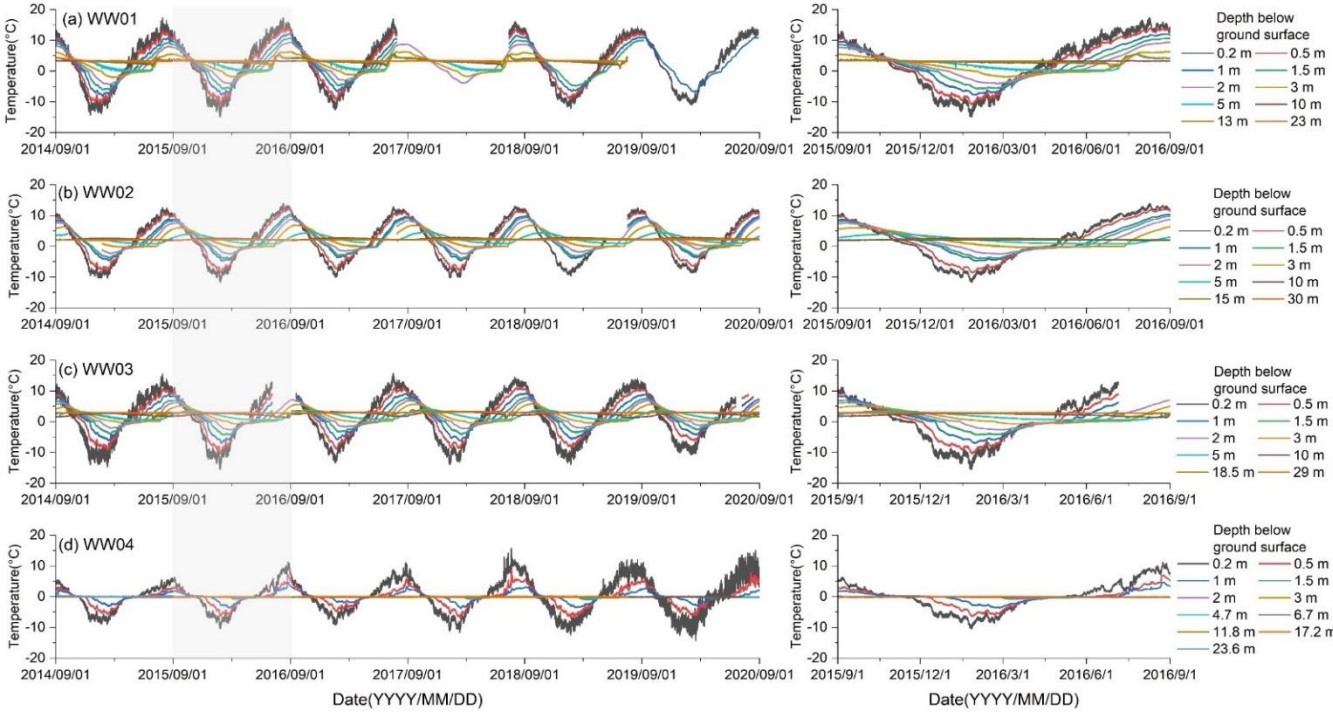

**Figure 7. The left panels show the dynamics of ground temperature at different depths over six consecutive years, while the right panels highlight the variation in ground temperature at different depths within a hydrological year.**

## 4 Collection and analysis of water samples

The following section presents the collection, preservation, and analysis of various water samples obtained in the study area from 2012 to 2017. Water samples were mainly collected in the following four stages: (1) January during the frozen period, (2) April and May during the thawing period, (3) July and August during the thawed period and (4) September during the refreezing period.

### 4.1 Sample collection and preservation

We collected samples of seven types of water in the study area, namely river water, precipitation, spring water, groundwater, soil water, glacier meltwater, and snow meltwater. The specific locations of

the sampling sites are shown in Fig. 1. All sample bottles were washed three times with filtered water and dried before use. During the collection of water samples (excluding precipitation and soil water samples), temperature (T), pH, dissolved oxygen (DO), electrical conductivity (EC), and oxidoreduction potential (ORP) were analyzed with a portable water quality analyzer (HQ40d, Hach, USA), which was calibrated for pH daily before use. The sampling procedure for different types of water is shown in Fig. 8. The river water and glacier meltwater samples were collected under natural flow conditions, and the stirring of riverbed sediments was carefully avoided. For the upwelling springs, water samples were collected at the center of the spring. The samples for the springs without upwelling were collected after the stagnant water was pumped out. For groundwater, at least 3-pore volumes water was pumped from the wells before sampling to ensure the old water was drained out.

A device made of stainless steel, as shown in Fig. 8e, was set up to collect snow meltwater samples in the field. The upper cover of the device was removed so that snow could fall into it. A small hole was cut in the bottom of the device and connected to a polyethylene pipe. When the temperature rises, the snow inside the device melts and the meltwater can slowly flow through the pipe into a polyethylene bottle at the other end of the pipe. In this way, the snowmelt water samples were collected. To collect precipitation, a device as shown in Fig. 8f was used. The circular funnel with a diameter of 14 cm in this device was made of polyethylene and was used to collect precipitation. Before each precipitation event, it was washed with ultrapure water to remove the fallout accumulated during the preceding dry period. Precipitation can pass through a polyester screen clamped in the top part of the funnel and into a polyethylene bottle at the bottom of the funnel. To minimize dry (dust) deposition, a ping-pong ball was set in the funnel. This device was held in place by a stainless steel cylinder, reinforced with stones around it.

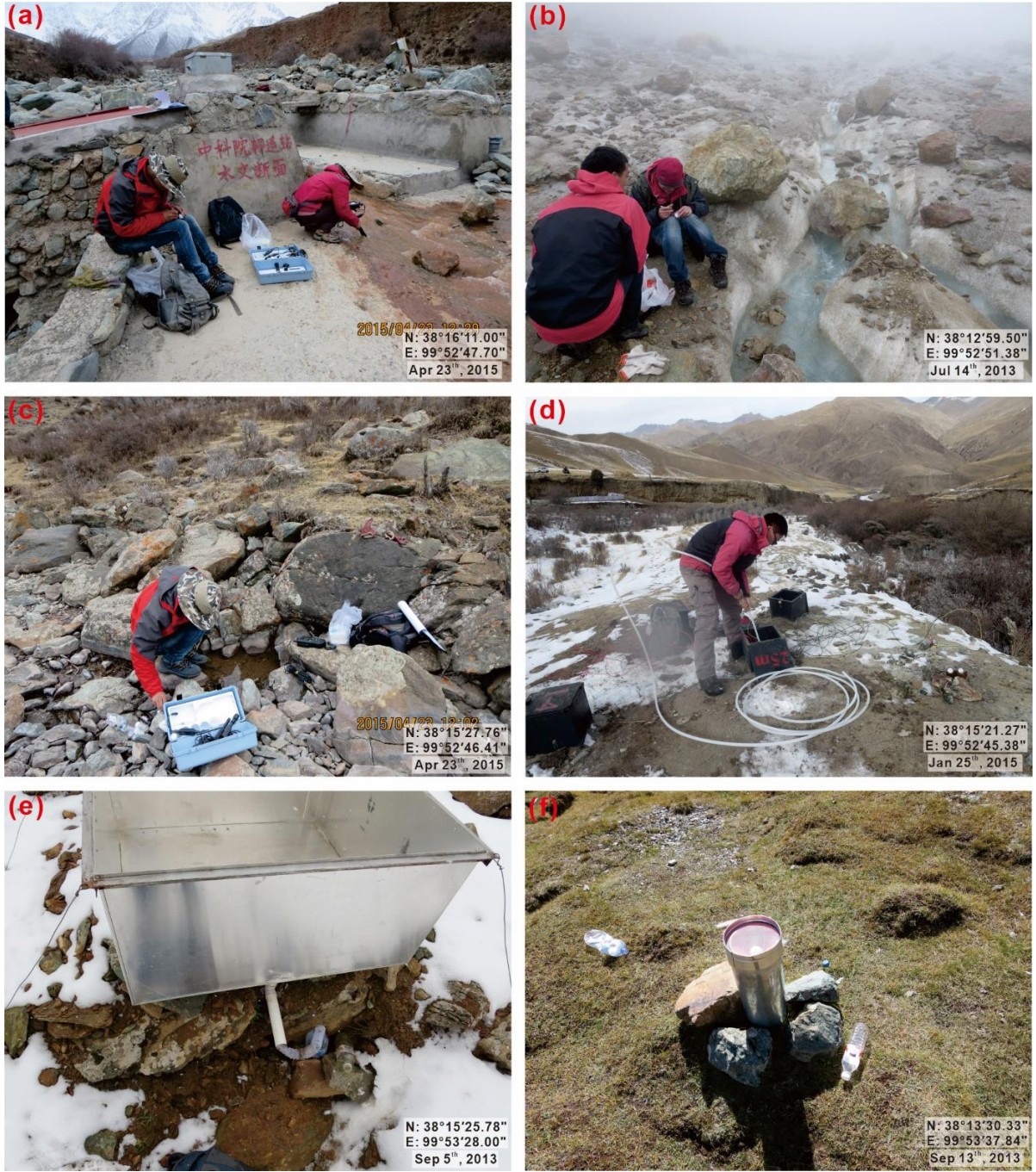

**Figure 8. Pictures showing procedures for sampling (a) river water, (b) glacier meltwater, (c) spring water, and (d) groundwater in the field. Devices for collecting snow meltwater and precipitation were shown in (e) and (f), respectively.**

Six subsets of water samples were collected in one sampling site to analyze different chemical and isotopic indicators. The treatment and preservation procedures for the six subsets were as follows: (1) the water samples for alkalinity titration were stored in 50-mL polyethylene bottles. Three replicates were collected for each sample without filtering, and alkalinity titration was performed within 24 hours of sampling; (2) the water samples for stable isotope ratio analysis were filtered with a 0.22-μm-pore-size membrane and stored in 2-mL clear glass bottles; (3) the water samples for radio isotope analysis were collected and stored in 1000-mL HDPE bottles; (4) the water samples for water chemistry analysis were filtered with a 0.22-μm-pore-size membrane and stored in 50-mL polyethylene bottles. Ultrapure $HNO_3$ was added to the bottled samples used for analyses of major cations and trace elements to ensure pH values $\leq 2$; (5) the water samples for DOC abundance analysis were filtered with a 0.45-μm-pore-size precombustion glass fiber filter membrane and stored in 40-mL brown threaded glass bottles. (6) the water samples for analysis of DIC and $\delta^{13}C$ value in DIC were filtered with a 0.22-μm-pore-size membrane and stored in 40-mL brown threaded glass bottles. All the above water samples were sealed in the bottles and stored at 4 °C.

Soil samples collected at different depths during the drilling of well group WW04 were sealed in 8-mL borate glass vials and stored at -10 °C. The specific borehole numbers and sample depths are listed in the worksheet named the "Soil water" of the second file in the dataset. The soil samples were then sent to the laboratory for water extraction using the cryogenic vacuum distillation technique (Sternberg et al., 1986; Smith et al., 1991) to prepare for subsequent stable isotope ratio measurements.

## 4.2 Water chemistry and isotope analysis of samples

### 4.2.1 Water chemistry analysis

The concentrations of anions ($Cl^-$, $NO_3^-$, and $SO_4^{2-}$) in water samples were determined by ion chromatography. The water samples collected in 2012 and 2013 were analyzed at the State Key Laboratory of Biogeology and Environmental Geology, China University of Geosciences using an ion

chromatograph (IC 761/813, Metrohm, Switzerland). The remaining samples collected from 2014–2017 were analyzed at the Laboratory of Basin Hydrology and Wetland Eco-restoration, China University of Geosciences using an ion chromatograph (Dionex ICS 1100, Thermo Elemental, USA). The concentrations of cations ($Ca^{2+}$, $K^+$, $Mg^{2+}$, and $Na^+$) as well as Si, and Sr, were measured at the Laboratory of Basin Hydrology and Wetland Eco-restoration, China University of Geosciences using an inductively coupled plasma atomic emission spectrometer (IRIS INTRE II XSP, Thermo Elemental, USA). The charge balance error of the measurements for all water samples was within ± 5%.

The DOC concentrations in water samples collected in 2012 and 2013 were measured at the laboratory of the Huazhong University of Science and Technology using a total organic carbon analyzer (Multi N/C 2100 TOC, Analytik Jena AG, Germany) with an analytical resolution of 0.001 ppb. The DIC concentrations in water samples collected in 2012 and 2013 were analyzed at the Third Institute of Oceanography, Ministry of Natural Resources using a stable isotope mass spectrometer (Delta V Advantage, Thermo Elemental, USA). The DOC and DIC concentrations in the remaining water samples collected from 2014–2017 were analyzed at the Laboratory of Basin Hydrology and Wetland Eco-restoration, China University of Geosciences using a total organic carbon analyzer (Aurora 1030W, OI, USA) with a precision of 50 μg/L.

### 4.2.2 Analysis of the stable isotopes of $^{13}C$, $^{18}O$, and $^{2}H$

The $^{13}C$ abundance in DIC in water samples collected in 2012 and 2013 was analyzed simultaneously while measuring the DIC concentrations using a stable isotope mass spectrometer (Delta V Advantage, Thermo Elemental, USA) at the Third Institute of Oceanography, Ministry of Natural Resources. The $^{13}C$ concentrations in the DIC of the groundwater samples from different boreholes (WW04–01, WW04–07, WW03–01, WW03–02, WW01–01, WW01–02, and WW01–03) and spring water samples (QW02, QW03, QW04, QW05, and QW08) collected in 2014 were measured using a wavelength-scanned cavity ring-down spectrometer (G2131-I, Picarro, USA). The $^{13}C$ in DIC results was expressed by the relative

abundance ($\delta$) of $^{13}$C in parts per thousand (‰), which was compared with Vienna PeeDee Belemnite (V-PDB). The analysis results of the water samples obtained in 2012 and 2013 are given in the corresponding dataset.

The $^2$H and $^{18}$O isotopic compositions of water samples collected in 2012 and 2013 were determined by a stable isotope ratio mass spectrometer (Finnigan MAT253, Thermo Elemental, USA) at the State Key Laboratory of Biogeology and Environmental Geology, China University of Geosciences. The $^2$H and $^{18}$O isotopic compositions of water samples collected in other periods were determined by an ultrahigh-precision liquid water isotope analyzer (L2130-I, Picarro, USA) at the School of Environmental Studies, China University of Geosciences. The analysis was repeated seven times for each sample, and the results of the first three replicates were ignored to eliminate the influence of the previous sample. Also, we established an internal calibration equation using standard samples (Vienna Standard Mean Ocean Water) to calibrate the analytical results. The results were expressed as relative abundance values ($\delta$) in parts per thousand (‰) compared to the standard samples. The analytical precision of the first method was 1.5‰ and 0.2‰ for $^2$H and $^{18}$O, respectively, and 0.5‰ and 0.1‰ for the second method, respectively.

### 4.2.3 Analysis of radioisotopes of $^3$H and $^{14}$C

The $^3$H and $^{14}$C concentrations were analyzed only for groundwater samples from different borehole (WW04–01, WW04–07, WW03–01, WW03–02, WW01–01, WW01–02, and WW01–03) and spring water samples (QW02, QW03, QW04, QW05 and QW08) collected in 2014. The water samples were concentrated by electrolysis and then analyzed for $^3$H using an ultra-low level scintillation spectrometer (Quantulus$^{TM}$ 1220, PerkinElmer, USA) at the Institute of Karst Geology, China Geological Survey. The $^3$H concentration was expressed in tritium units (TU) with a detection limit of ± 1 TU. The $^{14}$C concentration in the water samples was determined using a 3MV multi-element accelerator mass spectrometer (3MV Tandetron AMS, HVEE, Netherlands) at the Xi'an AMS Center in China. Before loading onto the instrument, the carbonate was removed from the samples by filtration through glass filter

paper under vacuum, and then 85% phosphoric acid was added to the samples.. Standard samples for $^{14}$C analysis were prepared according to the method described by Liu et al. (2019). The $^{14}$C analysis results were expressed as percent modern carbon (pmC) with an analytical precision of 2%.

The analytical methods used for the chemical and isotopic compositions of water samples are summarized in Table 5.

**Table 5. Summary of analytical methods for the chemical and isotopic compositions of water samples.**

| Indicators | Analytical instrument | Model of the instrument |
|---|---|---|
| T, pH, DO, EC, and ORP | Portable water quality analyzer | HQ40d, Hach, USA |
| Anions ($Cl^-$, $NO_3^-$, and $SO_4^{2-}$) | Ion chromatograph | IC 761/813, Metrohm, Switzerland |
| | | Dionex ICS 1100, Thermo Elemental, USA |
| $Ca^{2+}$, $K^+$, $Mg^{2+}$, $Na^+$, Si, and Sr | Inductively coupled plasma atomic emission spectrometer | IRIS INTRE II XSP, Thermo Elemental, USA |
| DOC | Total organic carbon analyzer | Multi N/C 2100 TOC, Analytik Jena AG, Germany |
| | | Aurora 1030W, OI, USA |
| DIC | Stable isotope mass spectrometer | Delta V Advantage, Thermo Elemental, USA |
| | Total organic carbon analyzer | Aurora 1030W, OI, USA |
| $^{13}$C | Stable isotope mass spectrometer | Delta V Advantage, Thermo Elemental, USA |
| | Wavelength-scanned cavity ring-down spectrometer | G2131-I, Picarro, USA |
| $^2$H and $^{18}$O | Stable isotope mass spectrometer | Finnigan MAT253, Thermo Elemental, USA |
| | Ultrahigh-precision liquid water isotope analyzer | L2130-I, Picarro, USA |
| $^3$H | Ultra-low level scintillation spectrometer | Quantulus$^{TM}$ 1220, PerkinElmer, USA |
| $^{14}$C | 3MV multi-element accelerator mass spectrometer | 3MV Tandetron AMS, HVEE, Netherlands |

## 5 General characteristics of water chemistry and isotopes in different waters

The mean concentrations of ions, Sr, and Si in river water, spring water, and groundwater are shown in Fig. 9. In general, the concentrations of ions, Sr, and Si were higher in groundwater and spring water than those in river water. In addition, the concentrations of ions, Sr, and Si in these waters were higher

during the frozen and refreezing periods than during the frozen and thawed periods. The spatiotemporal variations of water chemistry in different water bodies were affected by freeze-thaw processes. For example, the groundwater in permafrost zone exhibited dramatic variations in water chemistry.

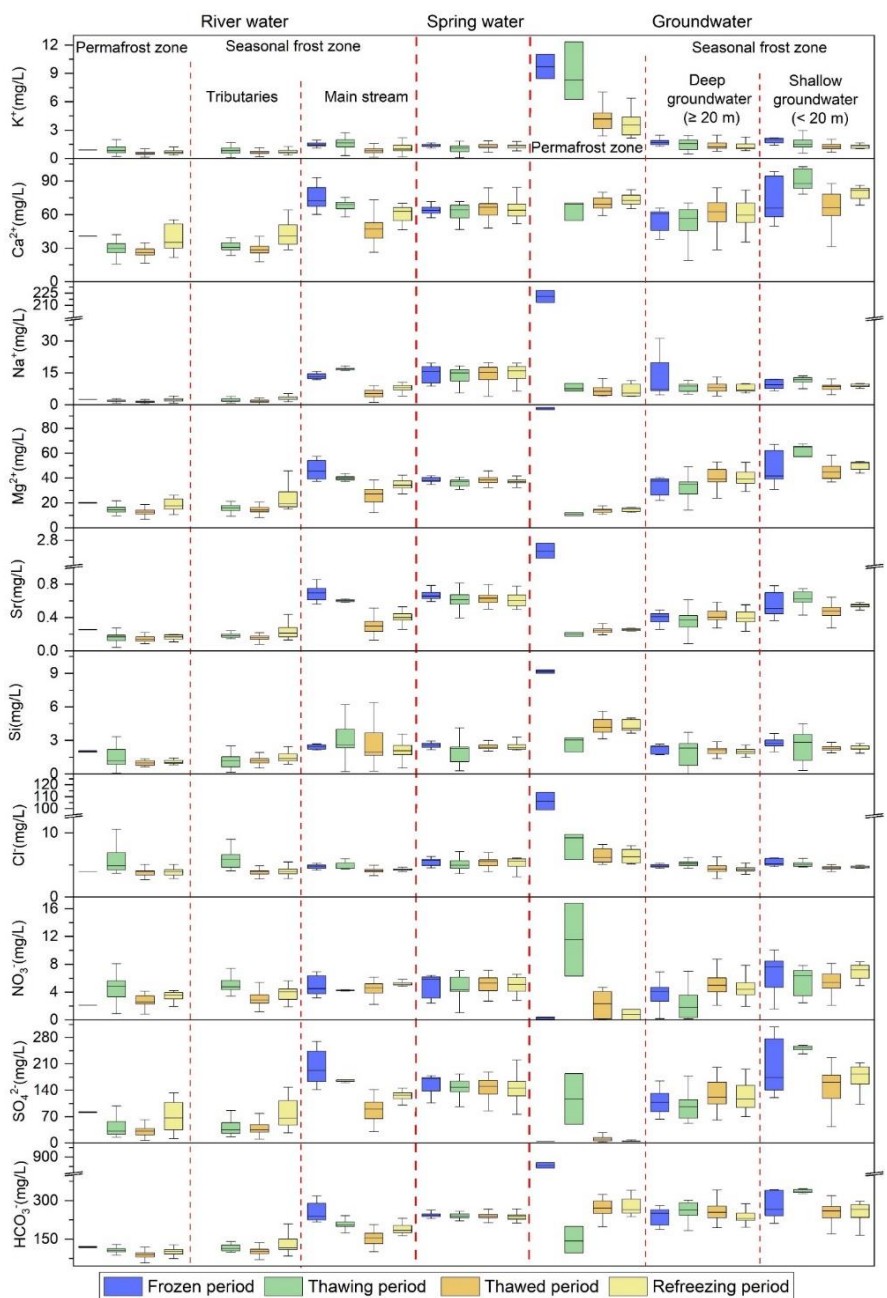

**Figure 9. Box plots showing the mean concentrations of ions, Sr, and Si in river water, spring water, and groundwater. The boxes denote the median and the 25th and 75th percentiles, while the whisker plots indicate the maximum and minimum, respectively.**

The concentrations of DOC and DIC in river water, spring water, and groundwater are shown in Fig. 10. The differences in DOC concentrations in both water bodies between the four periods were not significant, indicating a relatively stable DOC export. However, the DOC concentration in groundwater within the permafrost zone were much higher than in the other water bodies. In contrast, the DIC concentration showed a stronger temporospatial variation. For example, the DIC concentration of river water in the main stream was lower during the thawed period than during the frozen period, but higher than that in other river waters during all four periods.

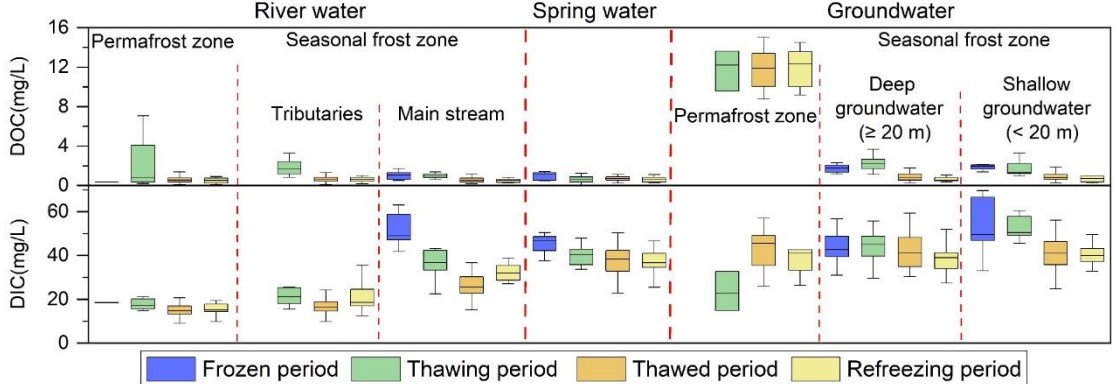

**Figure 10. Box plots showing the concentrations of DOC and DIC in river water, spring water, and groundwater. Boxes denote the median and the 25th and 75th percentiles, while whisker plots indicate maximum and minimum, respectively.**

The $\delta^{18}O$ and $\delta^2H$ relationship for different water samples and the local meteoric water line (LMWL) of $\delta^2H = 8.36\ \delta^{18}O + 18.97$ are shown in Fig. 11. The snow meltwater showed the largest variations in water stable isotopic compositions, with the $\delta^{18}O$ between -20.7‰ and -4.4‰ and the $\delta^2H$ between -155.0‰ and -13.1‰ (Fig. 11). In contrast, the isotopic compositions of glacier meltwater are less variable, with the $\delta^{18}O$ varying from -10.3‰ to-7.9‰ and the $\delta^2H$ varying from -61.9‰ to -39.3‰. The river water

showed a trend of decreasing variation along the flow in the isotopic compositions, with the $\delta^{18}O$ values of -12.3‰ to -7.5‰, -11.9‰ to 0.1‰, and -10.2‰ to -6.8‰ and the $\delta^2H$ values of -88.5‰ to -38.8‰, -83.1‰ to -14.7‰, and -62.7‰ to -41.2‰ for tributary water within the permafrost zone, tributary water within the seasonal frost zone and main stream water, respectively. Among the different types of groundwater, the spring water showed the least variation in water stable isotopic compositions, with the $\delta^{18}O$ ranging from -9.9‰ to -7.5‰ and the $\delta^2H$ ranging from -62.2‰ to -37.6‰; followed by the deep groundwater within the seasonal frost zone, with the $\delta^{18}O$ ranging from -7.9‰ to -10.4‰ and the $\delta^2H$ ranging from -63.3‰ to -43.4‰; the shallow groundwater within the seasonal frost zone had the $\delta^{18}O$ value ranging from -10.4‰ to -7.3‰ and the $\delta^2H$ value ranging from -64.7‰ to -8.1‰; and the suprapermafrost groundwater showed the greatest variation in water stable isotopic compositions, with the $\delta^{18}O$ value of -10.8‰ to -3.9‰ and the $\delta^2H$ value of -73.8‰ to -22.6‰.

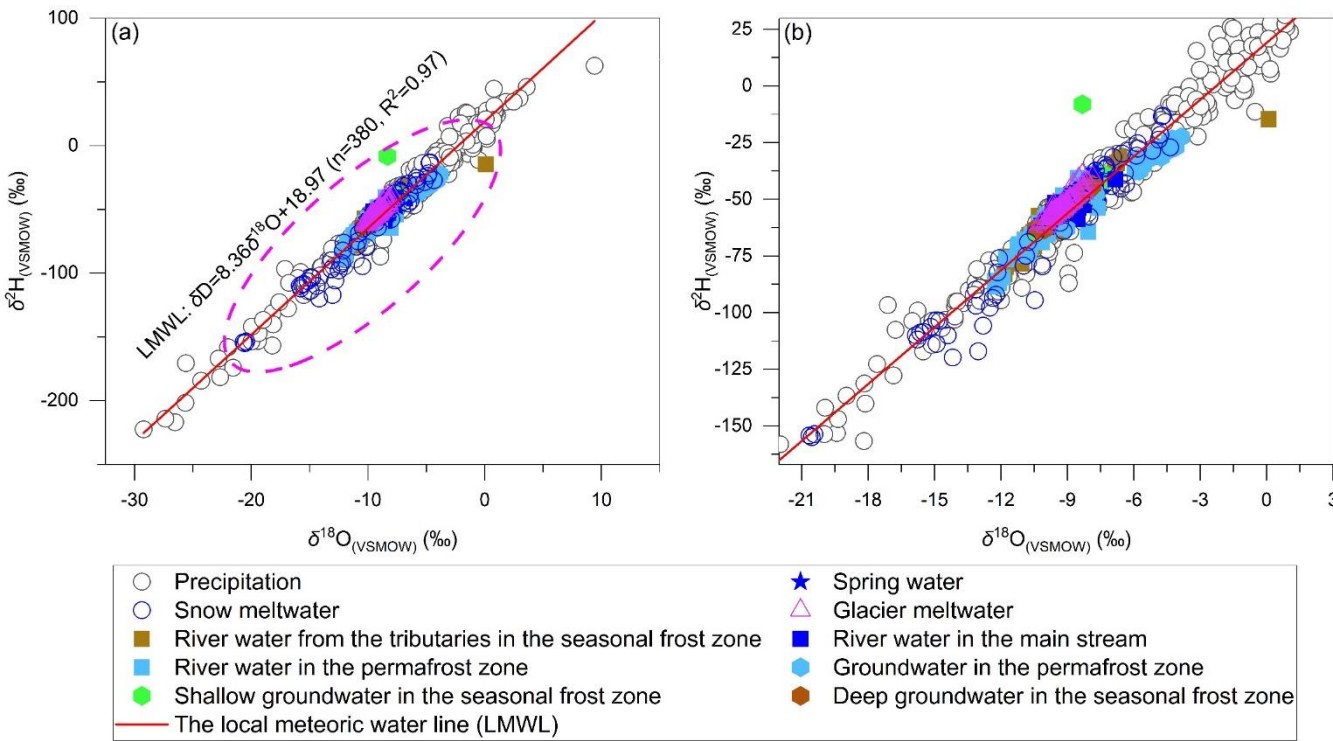

**Figure 11. (a) The $\delta^{18}O$ and $\delta^2H$ relationship for different types of water samples with (b) partially enlarged details of some water samples.**

The $^3$H concentrations in water samples ranged from 13.61 to 43.59 TU in Table 6. The $^{14}$C activity decreased from 76.43 and 96.34 pmC in permafrost zone to 49.60 and 44.38 pmC at the bottom of the piedmont plain, in contrast to a rise in values of $\delta^{13}$C from -16.77‰ to -5.92‰ (Table 6).

**Table 6. The $^{13}$C, $^3$H, and $^{14}$C concentrations in groundwater and spring water samples (Ma et al., 2017).**

| Sample No. | $\delta^{13}$C (‰) | | $^3$H (TU) | $^{14}$C activity (pmC) | |
|---|---|---|---|---|---|
| | $\delta^{13}$C | Error (1σ) | | $^{14}$C activity | Error (1σ) |
| WW04–01 (24.3-m borehole depth) | -16.77 | 0.51 | n.d. | 76.43 | 0.32 |
| WW04–07 (1.5-m borehole depth) | -13.60 | 0.57 | 15.11 | 96.34 | 0.31 |
| WW03–01 (30-m borehole depth) | -8.79 | 0.57 | 19.38 | 51.77 | 0.22 |
| WW03–02 (20-m borehole depth) | n.d. | n.d. | 16.22 | n.d. | n.d. |
| WW01–01 (25-m borehole depth) | -5.92 | 0.53 | 16.95 | 49.60 | 0.18 |
| WW01–02 (15-m borehole depth) | n.d. | n.d. | 24.18 | 44.38 | n.d. |
| WW01–03 (10-m borehole depth) | n.d. | n.d. | 16.20 | n.d. | n.d. |
| QW02 (spring) | n.d. | n.d. | 27.83 | n.d. | n.d. |
| QW03 (spring) | n.d. | n.d. | 13.84 | n.d. | n.d. |
| QW04 (spring) | n.d. | n.d. | 13.61 | 43.05 | 0.19 |
| QW05 (spring) | -5.09 | 0.7 | 43.59 | n.d. | n.d. |
| QW08 (spring) | n.d. | n.d. | 18.58 | n.d. | n.d. |

"n.d." means not determined.

## 6 Data availability

All data mentioned in this paper are available at https://doi.org/10.5281/zenodo.6296057 (Ma et al., 435 2021b). The raw data were divided into two files. The first file contains the monitoring data, including groundwater levels and ground temperatures. The second file contains the measurement results of the samples with their numbers, sampling locations (latitude and longitude coordinates), and elevations.

## 7 Conclusions

This study presents a systematic hydrogeological and hydrochemical monitoring scheme for an

alpine catchment located in the headwaters of the Heihe River, which is underlain by permafrost and seasonal frost. A total of 22 boreholes in four well groups were drilled, and the groundwater levels in eight of the boreholes and ground temperatures were monitored at 30-minute intervals for six consecutive years. Samples of sediment, river water, precipitation, spring water, groundwater from boreholes, soil water, glacier meltwater, and snow meltwater were collected during the frozen (January), thawing (April and May), thawed (July and August), and refreezing (September) periods from 2012 to 2017. The concentrations of anions ($Cl^-$, $NO_3^-$, and $SO_4^{2-}$), major cations ($Ca^{2+}$, $K^+$, $Mg^{2+}$, and $Na^+$), Si, Sr, DOC and DIC and the compositions of stable isotopes ($^{13}C$ in DIC and $^2H$ and $^{18}O$ in water) and radioisotopes($^3H$ and $^{14}C$) in the water samples were measured.

This study is among the few attempts in alpine regions to obtain groundwater level, temperature, chemistry, and isotopic data representing the subsurface in-situ environment based on direct sampling from drilled boreholes. These data, combined with other measurements such as river discharge, precipitation, water chemistry, and isotopic composition of glacier and snow meltwater, can be used to explore different kinds of groundwater-related issues in areas dominated by permafrost and seasonal frost, including but not limited to (1) the effect of soil freeze-thaw processes on groundwater flow and groundwater-surface water interactions; (2) the interplay between permafrost degradation and groundwater flow changes; and (3) the water quality in alpine catchments under the influence of seasonal freeze-thaw processes of soils and permafrost degradation. Based on this dataset, the following efforts should be made in the future to better understand the coupled hydrobiogeochemical cycles in alpine catchments under the impact of climate change: (1) coupling with other monitoring systems, such as meteorological monitoring systems, remote sensing monitoring systems, and hydrological monitoring systems, to obtain data on climatic characteristics, permafrost distribution and dynamics,, ecosystem status, and hydrological regimes; (2) developing integrated flow-thermal-solute transport models to explore the impacts of seasonal freeze-thaw processes in soils and permafrost degradation on groundwater flow, groundwater-surface water interaction, element cycling, and water quality; and (3) assessing the

impacts of groundwater flow evolutions on regional biogeochemical cycles under climate change.

**Author contributions.** RM and ZS designed the whole monitoring system. ZP, YH, QC, MG, SW, JB, XL, YP, and LZ performed the fieldwork in the Hulugou catchment and the pre-treatment work of samples. ZP and RM collated the dataset. All co-authors participated in writing and revising of the paper.

**Competing interests.** The authors declare that they have no conflicts of interest.

**Acknowledgments.** We sincerely thank all the staff of the Qilian Alpine Ecology and Hydrology Research Station, Northwest Institute of Eco-Environment and Resources, Chinese Academy of Sciences.

**Financial support.** This work was financially supported by the Strategic Priority Research Program of the Chinese Academy of Sciences (No. XDA20100103), the National Natural Science Foundation of China (No. 41772270), and the Natural Sciences Foundation of Hubei Province of China (No. 2019CFA013).

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
