# Peer review of "Integrated hydrogeological and hydrogeochemical dataset of an alpine catchment in the northern Qinghai-Tibet Plateau"

_Earth System Science Data, 2021_

## Referee Comment (RC1)

**Review**

It's my pleasure to review "Datasets for research on groundwater flow and its interactions with surface water in an alpine catchment on the northeastern Tibetan Plateau, China" by Pan et al. This is an unique and very important dataset fur understanding the permafrost hydrologic process on the Tibetan Plateau. The manuscript is generally well orgnized and written. The manuscript can be accepted after addressing my following comments.

**Main comments**

1. In the manuscript, the authors mentioned the precipitation, temperature, and streamflow data in the study area (p.5 l.108) but gave only a URL (http://hhsy.casnw.net). How to download the data is not mentioned. Given the importance of the precipitation, temperature, and streamflow data to the use of datasets in this manuscript, I strongly recommended adding meteorological stations and hydrological monitoring points in the Study Area section and explaining how to apply for and download these data.

2. As a descriptive manuscript related to the field monitoring data, it should present the details about the study area as much as possible. However, the study area description in the manuscript, such as permafrost and planation surface, is too broad. It does not promote my understanding of the conditions at these sites. Can the authors provide some pictures of each typical landform, the well-group layout, and the core lithology of the borehole? Furthermore, please add a geological map of the study area.

3. In the alpine area with extreme weather conditions, the sensor's accuracy is especially essential in the quality assurance of field monitoring data. And to gain

this, a priority is to conduct systemic sensor calibration. However, no information about the sensor calibration can be found in the manuscript. Therefore, I highly recommend reinforcing the information about the processes and results of sensor calibrating in the manuscript.

4. Similarly, detailed information about the processes of water sampling is vital to evaluate the quality of hydrochemical and isotopic data. Unfortunately, I cannot find any information regarding this. So, it is suggested to provide detailed information (better with some photos) about sampling processes of different water reservoirs, including precipitation, glacier meltwater, groundwater, etc. In addition, the accuracy controlling of analytical results needs to be explained.

**Minor comments**

These are my minor comments, in order of appearance:

1. p.7 l.150: Please explain the data source for the permafrost range in figures 1(b) and 1(c) and the source and resolution of elevation data in Figure 1(c)? In addition, the phrase 'spring water' was used in the manuscript, while the word 'spring' was used in Figure 1(c). Please keep consistent.

2. p.12 l.203: What does '/' mean in Table 2? Is it the same as 0%? Please explain clearly.

3. p.13 l.210: The interval of temperature monitoring seems to have been forgotten to mention.

4. p.14 l.224: In Figure 3, the lines used to represent the groundwater level are broken. Is it due to missing data or something else? There is a similar problem in Figure 4 (p.15 l.233). It is recommended to tabulate the available data in the manuscript.

**5. Section 4.2, 4.3, 4.4, and 4.3: These sections involve various analysis methods of water samples. A table is suggested to add to make a summary about these methods.**

**Technical comments**

**1. p.7 l.150: Since many monitoring and sampling sites belonging to different types were involved, it is recommended to rearrange Figure 1 to obtain a better visual effect.**

**2. p.10 l.152: 'Table 1 (continued)' should be changed to 'Table 1. (continued)'.**

**3. p.13 l.209: Please check the resolution of the pressure sensor (HOBO U20–001–02, ONSET, USA). The resolution should be 0.41 cm, not 0.21 cm.**

**4. p.13 l.210: The wire length of the temperature sensor (HOBO S–TMB–M0017, ONSET, USA) is only 17 m. Just out of curiosity: how to measure the temperature to a depth of 30 m?**

**5. p.13 l.213: Please check the resolution of the temperature sensor (HOBO S–TMB–M0017, ONSET, USA). The resolution should be ± 0.03 °C, not 0.03 °C.**

**6. p.14 l.224 and p.15 l.233: It is tough to see the differences described in your text from Figures 3 and 4 because the x-axis covers a range as long as six years. Please consider adding another plot or subplot with a shorter time span.**

**7. p.16 l.260: Please double-check the pore diameter of the filter membrane used for DOC water sampling. It should be 0.7 μm.**

**8. p.18 l.291: 'ppb' is generally not used to represent the unit of measurement precision. It is suggested to be replaced by '$u$g/L'.**

**9. p.18 l.300: A space is missing in 'of' and '$^{13}$C'.**

**10. p.28 l.590 and p.28 l.608: The URLs of the two references cannot be opened. Please double-check.**

**The dataset**

**These comments concern only the dataset itself given at**
**https://doi.org/10.5281/zenodo.5184470**

**1. There seems to be a language problem with the title of the dataset. Please correct it.**

**2. Please add elevation data for each site in the datasets.**

**3. Please simplify the analysis results. For example, the DOC concentration in precipitation was not measured using the total organic carbon analyzer (Multi N/C 2100 TOC, Analytik Jena AG, Germany). Thus this column can be deleted from the datasets.**

**4. It seems that some groundwater level data were missing. Please explain the reasons for the data missing (please refer to Minor comment #3).**

---

## Referee Comment (RC2)

The manuscript entitled "Datasets for research on groundwater flow and its interactions with surface water in an alpine catchment on the northeastern Tibetan Plateau, China" investigates the alpine groundwater flow and its interactions with surface water through a series of field experiments. Groundwater is critical for alpine regions but is often overlooked and in particular with respect to prospected degrading permafrost and glaciers under global climatic warming. The understanding for subsurface flow in alpine regions has long been limited because of the lack of observed data to explore. Thus, the topic and research of this study fit well the scope of ESSD and merits publication in this journal. However, there are some major issues and specific logical flaws in this stage.

**Major issues**
1. Although the ESSD aims to provide high quality datasets for the earth science field, it is still a scientific journal to present the new results, knowledge and understanding based on obtained dataset, rather than simply describing the measurements of datasets. The manuscript in this stage seems like a field experiment report, but lack the critical analysis for datasets, the knowledge indicated from datasets and understanding implicated from datasets.
   For example, the groundwater level variations from four well groups indicate that deeper depth has the larger variations (over 15 m) of groundwater head (Figure 3), which implicates the mountainous groundwater level particular in high elevation would be very sensitive to changes in land surface or hydrogeology structure (thawing permafrost will reshape the hydraulic connectivity). All of these observations are worthy of in-depth analysis and discussion.
   **Therefore, I would like to suggest restructuring the section 3 and 4, presenting the data with a clear result analysis and discussion part, to demonstrate how these data help improving understanding for surface-groundwater interactions of Tibet Plateau, and what insights to provide for other alpine regions.**

2. The authors give detailed descriptions for observed process, but lack a clear scientific objective for experimental design. Readers will be wondering, such as, why they put wells in those four locations, why they collected samples from western tributary…
   Thus, a clear objective corresponding to measurement should be need, which will strongly demonstrate the significance of the study and datasets.

**Specific issues:**
1. Title: "research on groundwater flow and its interactions with surface water" is so general that easy to lose readers who hope to find relevant datasets. Moreover, the manuscript lacks the in-depth analysis of how these data indicate the groundwater flow and surface-groundwater interaction. So I would like to suggest a direct and clear title, such as "Dataset for alpine groundwater levels and hydrogeochemistry in northern Tibetan Plateau".

2. Lines 58-59: Actually there are some groundwater studies focused on TP, and need to be included.
   *Yao, Y., et al. (2017). "What Controls the Partitioning between Baseflow and Mountain Block Recharge in the Qinghai-Tibet Plateau?" Geophysical Research Letters 44(16): 8352-8358.*
   *Yao, Y., et al. (2021). "Role of Groundwater in Sustaining Northern Himalayan Rivers." Geophysical Research Letters 48(10).*

3. Line 94: "Groundwater level and ground temperature changes are also explained". Explain what? Maybe it is better to use "show".

4. Line 100: Since the whole manuscript did not mentioned the significance of this study for the Heihe River, I would like to suggest highlighting that this is a typical case for permafrost regions or headwater areas of TP.

5. Line 104: should be "annual averaged precipitation".

6. Line 107: Since the daily discharge is much highly varied, I would like to convert the volume unit of ($m^3$/day) to depth (mm/year).

7. Line 121: Should be "good hydraulic connectivity".

8. Line 124: Since the precipitation involves the snow, this should be revised as "the aquifer is recharged by rainfall, melt water from glaciers and snow".

9. Line 126: How low of the vegetation coverage? This should provide a percentage value at least. And this sentence is inconsistent with the following descriptions on vegetation coverage in line 135 (shrubs) and 147 (meadows). All these three parts should be combined and presented in consistency and with a clear percentage.

10. Line 128-129: how about the active layer thickness, 2 m?

11. Line 140: Is this unconfined aquifer?

12. Figure 1: I would like to suggest adding one or two photos to show your field experiments.

13. Line 155: "Well groups" will make readers misunderstanding there are multiple wells in a group. Direct using well would be better, and indicate one well includes multiple boreholes in different depth.

14. Line 181: Is this clay layer the top of the confined aquifers?

15. Line 218-220: Should give a detailed discussion for WW04, because it is the only well in the permafrost area.

16. Figure 3: The caption should note this is "daily variations".

17. Section 4.2 and 4.3: Any results and information we obtained from these part of datasets?

---

## Author Comment (AC1)

**Response to Reviewer 1:**

We would like to thank you for your positive comments and helpful suggestions. We revised our manuscript accordingly and are providing below a point-by-point response to each comment.

**Main comments**

1. In the manuscript, the authors mentioned the precipitation, temperature, and streamflow data in the study area (p.5 1.108) but gave only a URL (http://hhsy.casnw.net). How to download the data is not mentioned. Given the importance of the precipitation, temperature, and streamflow data to the use of datasets in this manuscript, I strongly recommended adding meteorological stations and hydrological monitoring points in the Study Area section and explaining how to apply for and download these data.

Response: To address this comment, we have added the locations of meteorological and stream gauging stations in Fig. 1, and also described how to access these data in detail in the revised manuscript.

The sentences read now as:

"The hydrometeorological monitoring network in the study area is composed of five automatic meteorological stations and one stream gauging station (Fig. 1), which are maintained and operated by the Qilian Alpine Ecology and Hydrology Research Station, Northwest Institute of Eco-Environment and Resources, Chinese Academy of Sciences (http://hhsy.casnw.net). Researchers with reasonable request for the precipitation, air temperature and streamflow data in the study area can apply for via email. The website for the specific contact information is: http://hhsy.casnw.net/lxwm/index.shtml."

---

## Author Response (AR1)

Dear Editor,

We are submitting our revised manuscript essd-2021-273 "Multi-year dataset for groundwater level, temperature, and chemical and isotopic compositions of different water bodies in an alpine catchment on the northeastern Qinghai-Tibet Plateau, China" to your prestigious journal.

We would like to thank two reviewers for their valuable comments, which have improved the presentation and quality of this work. We have revised our manuscript according to the reviewers' comments. A point-by-point response to each of the reviewers' comments is also provided in the reply letter.

We hope that you find this revised manuscript is suitable for publication in your journal.

Thank you very much for handling our manuscript.

Sincerely,

Rui Ma on behalf of all authors
* * *
**Response to Reviewer 1:**

**Main comments**

1. In the manuscript, the authors mentioned the precipitation, temperature, and streamflow data in the study area (p.5 l.108) but gave only a URL (http://hhsy.casnw.net). How to download the data is not mentioned. Given the importance of the precipitation, temperature, and streamflow data to the use of datasets in this manuscript, I strongly recommended adding meteorological stations and hydrological monitoring points in the Study Area section and explaining how to apply for and download these data.

Response: To address this comment, we have added the locations of meteorological and stream gauging stations in Fig. 1, and also described how to access these data in detail in the revised manuscript.

The sentences read now as:

*"The hydrometeorological monitoring network of the catchment is composed of five automatic meteorological stations and one stream gauging station (Fig. 1), which are maintained and operated by the Qilian Alpine Ecology and Hydrology Research Station, Northwest Institute of Eco-Environment and Resources, Chinese Academy of Sciences (http://hhsy.casnw.net). Researchers with a reasonable need for the precipitation, air temperature, and runoff data for the study area may request them by email. The website for the specific contact information is http://hhsy.casnw.net/lxwm/index.shtml."*

[Figure]

Figure 1. (a) The location of the study area in the headwater regions of the Heihe River; and (b) the Hulugou catchment showing the topography and the monitoring and sampling sites. The permafrost distribution and elevation data were obtained from the National Tibetan Plateau Data Center (http://data.tpdc.ac.cn), and the resolution of elevation data is 5 meters.

2. As a descriptive manuscript related to the field monitoring data, it should present the details about the study area as much as possible. However, the study area description in the manuscript, such as permafrost and planation surface, is too broad. It does not promote my understanding of the conditions at these sites. Can the authors provide some pictures of each typical landform, the well group layout, and the core lithology of the borehole? Furthermore, please add a geological map of the study area.

Response: We now have added the pictures showing each typical landform, the well group layout, and the core lithology of the borehole, as shown in Fig. 2, Fig. 3, and Fig. 4. In addition, we have put the geological map in the supplementary file.

[Figure]

Figure 2. Pictures showing (a) the glaciers in the south part of the study area, (b) the moraine sediments in the periglacial zone, (c) the planation surface in the permafrost zone, and (d) the piedmont alluvial plain in the seasonal frost zone.

[Figure]

Figure 3. Pictures showing the field scenes for (a) the location and layout of the four well groups in the study area, (b) the layout of wells in the well group WW01, (c) the layout of wells in the well group WW02, (d) the layout of wells in the well group WW03, and (e) the layout of wells in the well group WW04.

[Figure]

Figure 4. Pictures showing the lithology of cores at different depths belowground from the well groups (a) WW01, (b) WW02, (c) WW03, and (d) WW04.

The geological map of the study area.

[Figure]

Figure S1. (a) Geologic map of the study area; (b) a geological cross section (Modified from Liu, 2013; Ma et al., 2017; Chang, 2019).

References

Chang, Q.: Water Sources of Stream Runoff in Alpine region and their Seasonal Variations: onA Case Study of Hulugou Catchment in the Headwaters of the Heihe River, Ph.D. , School of Environmental Studies, China University of Geosciences Wuhan, 158 pp., https://doi.org/10.27492/d.cnki.gzdzu.2019.000112, 2019.

Liu, Y.: Using hydrochemical and isotope tracers analing to delineate hydrologic process in cold alpine watershed in rainy season, Ph.D., School of Environmental Studies, China University of Geosciences Wuhan, 104 pp., 2013.

Ma, R., Sun, Z., Hu, Y., Chang, Q., Wang, S., Xing, W., and Ge, M.: Hydrological connectivity from glaciers to rivers in the Qinghai–Tibet Plateau: roles of suprapermafrost and subpermafrost groundwater, Hydrol. Earth Syst. Sci., 21, 4803-4823, https://doi.org/10.5194/hess-21-4803-2017, 2017.

3. In the alpine area with extreme weather conditions, the sensor's accuracy is especially essential in the quality assurance of field monitoring data. And to gain this, a priority is to conduct systemic sensor calibration. However, no information about the sensor calibration can be found in the manuscript. Therefore, I highly recommend reinforcing the information about the processes and results of sensor calibrating in the manuscript.

Response: All temperature sensors were calibrated in the laboratory before use. We placed each temperature sensor in water under eight different temperatures (-40 °C, -30 °C, -20 °C, 0 °C, 10 °C, 20 °C, 30 °C, and 40 °C) and the temperatures were measured using the sensors. The slopes and the correlation coefficients between measured vs. actual values for all sensors were in the range of 0.998–1.003.

Similarly, all pressure sensors were placed under the condition of the nine different pressures (10 kpa, 50 kpa, 100 kpa, 150 kpa, 200 kpa, 250 kpa, 300 kpa, 350kpa, and 399 kpa), and the pressures were measured by these sensors. The slopes and the correlation coefficients between measured vs. actual values for all sensors were in the range of 0.999–1.001.

We have added the description of sensor calibration in the supplementary file of the revised manuscript.

4. Similarly, detailed information about the processes of water sampling is vital to evaluate the quality of hydrochemical and isotopic data. Unfortunately, I cannot find any information regarding this. So, it is suggested to provide detailed information (better with some photos) about sampling processes of different water reservoirs, including precipitation, glacier meltwater, groundwater, etc. In addition, the accuracy controlling of analytical results needs to be explained.

Response: We have added some photos showing water sampling process in the revised manuscript, as shown in Fig. 5. The river water and glacier meltwater samples were collected under natural flow conditions, and the stirring of riverbed sediments was carefully avoided. For the upwelling spring, water samples were collected at the center of the spring. The samples for the springs without upwelling were collected after the stagnant water was pumped out. For groundwater, at least 3-pipe volumes water was pumped from the wells before sampling to ensure the old water was drained out.

A device made of stainless steel, as shown in Fig. 5e, was set up to collect snow meltwater samples in the field. The upper cover of the device was removed so that snow could fall into it. A small hole was cut in the bottom of the device and connected to a polyethylene pipe. When the temperature rises, the snow inside the device melts and the meltwater can slowly flow through the pipe into a polyethylene bottle at the other end of the pipe. In this way, the snowmelt water samples were collected. To collect precipitation, a device as shown in Fig. 5f was used. The circular funnel with a diameter of 14 cm in this device was made of polyethylene and was used to collect precipitation. Before each precipitation event, it was washed with ultrapure water to remove the fallout accumulated during the preceding dry period. Precipitation can pass through a polyester screen clamped in the top part of the funnel and into a polyethylene bottle at the bottom of the funnel. To minimize dry (dust) deposition, a ping-pong ball was set in the funnel. This device was held in place by a stainless steel cylinder, reinforced with stones around it.

The sampling processes for different water reservoirs were described in the revised manuscript.

During measurement of major ions, DOC, DIC in the laboratory of our university, the standard curve was carefully prepared and the stability of instrument was tested before measuring the samples. The correlation coefficients of standard curves were greater or equal to 0.9999. The quality control samples (the chemical or isotopic concentrations of water were known) were tested at every five-eight measured samples to check the

data quality. The deviation between measured and true (reference) values for the quality control samples was less than 5 %, so as to ensure the stability of instrument. When the measurements of major ions concentrations were finished, we also calculated the charge balance for the water samples and the errors were within ± 5%. Since some samples were sent to the professional analysis institutions (including the laboratory of the Huazhong University of Science and Technology, and the Third Institute of Oceanography, Ministry of Natural Resources), which have been certified by professional laboratory qualification, the data quality should be ensured. We have added the content about the accuracy controlling of analytical results in the revised manuscript.

[Figure]

Figure 5. Pictures showing procedures for sampling (a) river water, (b) glacier meltwater, (c) spring water, and (d) groundwater in the field. Devices for collecting snow meltwater and precipitation were shown in (e) and (f), respectively.

**Minor comments**

These are my minor comments, in order of appearance:

1. p.7 l.150: Please explain the data source for the permafrost range in figures 1(b) and 1(c) and the source and resolution of elevation data in Figure 1(c)? In addition, the phrase 'spring water' was used in the manuscript, while the word 'spring' was used in Figure 1(c). Please keep consistent.

Response: The data source for the permafrost range and the elevation in Figures 1(b) and 1(c) are obtained from the National Tibetan Plateau Data Center at website http://data.tpdc.ac.cn. The resolution of elevation data is 5 meters in Figure 1(c). We have modified the caption of Figure 1 in the revised manuscript. Please refer to the response to Main comments #1.

2. p.12 l.203: What does '/' mean in Table 2? Is it the same as 0%? Please explain clearly.

Response: The '/' symbol in Table 2 is same as 0%. To be consistent, we have changed the '/' symbol to 0. Please refer to new Table 2 in the revised manuscript.

3. p.13 l.210: The interval of temperature monitoring seems to have been forgotten to mention.

Response: We have added the interval of temperature monitoring (30-minute intervals) in the revised manuscript.

4. p.14 l.224: In Figure 3, the lines used to represent the groundwater level are broken. Is it due to missing data or something else? There is a similar problem in Figure 4 (p.15 l.233). It is recommended to tabulate the available data in the manuscript.

Response: The main reason for the broken lines in the hydrographs is that some groundwater level data are missing because that the groundwater tables were deeper than wells screen depths during cold periods. The missing data for temperature is due to some problems of the sensors, such as damage or power failure. We have tabulated the available data in the revised manuscript, as shown in Table 1 and Table 2.

Table 1. The periods with available groundwater level data.

| Well group no. | Borehole depth (m) | Periods with available data |
|---|---|---|
| WW01 | 5 | 2014/08–2014/12, 2015/07–2015/12, 2016/07–2016/12, 2018/08–2018/12, 2019/07–2019/12 |
| | 10 | 2014/08–2015/01, 2015/06–2016/01, 2016/06–2017/01, 2017/06–2018/01, 2018/06–2019/01, 2019/06–2020/01, 2020/06–2020/08 |
| | 15 | 2014/08–2015/02, 2015/06–2016/02, 2016/06–2017/02, 2017/06–2018/02, 2018/06–2019/01, 2019/06–2019/09 |
| | 25 | 2014/08–2015/02, 2015/06–2015/07, 2015/09–2019/08 |
| WW03 | 20 | 2014/08–2014/10, 2015/04–2015/06, 2015/07–2020/08 |
| | 30 | 2014/08–2014/12, 2015/04–2020/08 |
| WW04 | 1.5 | 2014/09–2015/01, 2015/07–2016/01, 2016/09–2017/01, 2017/04–2017/07, 2018/07–2019/12, 2019/08–2019/12, 2020/07–2020/08 |
| | 24.3 | 2015/08–2016/07 |

Table 2. The periods with available ground temperature data.

| Well group no. | Depths (m) | Periods with available data |
|---|---|---|
| WW01 | 0.2, 1 | 2014/09–2017/07, 2018/07–2020/07 |
| | 0.5, 1.5 | 2014/09–2017/07, 2018/07–2019/09 |
| | 2, 3, 5 | 2014/09–2018/11 |
| | 10, 13, 23 | 2014/09–2019/07 |
| WW02 | 0.2, 1, 1.5 | 2014/09–2019/05, 2019/07–2020/08 |
| | 0.5 | 2014/09–2018/09, 2019/07–2020/08 |
| | 2, 3, 5 | 2014/09–2014/10, 2015/01–2017/04, 2017/08–2020/08 |
| | 10, 15, 30 | 2014/09–2020/08 |
| WW03 | 0.2, 0.5, 1, 1.5 | 2014/09–2016/07, 2016/9–2020/06, 2020/07–2020/08 |
| | 2, 3, 5, 10, 18.5, 29 | 2014/09–2020/08 |
| WW04 | 0.2, 0.5, 1, 1.5 | 2014/09–2020/08 |
| | 2, 3, 4.7, 6.7 | 2014/09–2019/04, 2019/07–2020/06 |
| | 11.8, 17.2 | 2014/09–2019/07 |
| | 23.6 | 2014/09–2015/09 |

5. Section 4.2, 4.3, 4.4, and 4.3: These sections involve various analysis methods of water samples. A table is suggested to add to make a summary about these methods.

Response: We have added Table 3 to summarize the analysis methods in the revised manuscript.

Table 3. Summary of analytical methods for the chemical and isotopic compositions of water samples.

| Indicators | Analytical instrument | Model of the instrument |
|---|---|---|
| T, pH, DO, EC, and ORP | Portable water quality analyzer | HQ40d, Hach, USA |
| Anions ($Cl^-$, $NO_3^-$, and $SO_4^{2-}$) | Ion chromatograph | IC 761/813, Metrohm, Switzerland |
| | | Dionex ICS 1100, Thermo Elemental, USA |
| $Ca^{2+}$, $K^+$, $Mg^{2+}$, $Na^+$, Si, and Sr | Inductively coupled plasma atomic emission spectrometer | IRIS INTRE II XSP, Thermo Elemental, USA |
| DOC | Total organic carbon analyzer | Multi N/C 2100 TOC, Analytik Jena AG, Germany |
| | | Aurora 1030W, OI, USA |
| DIC | Stable isotope mass spectrometer | Delta V Advantage, Thermo Elemental, USA |
| | Total organic carbon analyzer | Aurora 1030W, OI, USA |
| $^{13}C$ | Stable isotope mass spectrometer | Delta V Advantage, Thermo Elemental, USA |
| | Wavelength-scanned cavity ring-down spectrometer | G2131-I, Picarro, USA |
| $^2H$ and $^{18}O$ | Stable isotope mass spectrometer | Finnigan MAT253, Thermo Elemental, USA |
| | Ultrahigh-precision liquid water isotope analyzer | L2130-I, Picarro, USA |
| $^3H$ | Ultra-low level scintillation spectrometer | Quantulus$^{TM}$ 1220, PerkinElmer, USA |
| $^{14}C$ | 3MV multi-element accelerator mass spectrometer | 3MV Tandetron AMS, HVEE, Netherlands |

**Technical comments**

1. p.7 l.150: Since many monitoring and sampling sites belonging to different types were involved, it is recommended to rearrange Figure 1 to obtain a better visual effect.

Response: We have rearranged the layout of Figure 1 in the revised manuscript. Please also refer to the response to the Main comments #1.

2. p.10 l.152: 'Table 1 (continued)' should be changed to 'Table 1. (continued)'.

Response: Change was made as suggested.

3. p.13 l.209: Please check the resolution of the pressure sensor (HOBO U20–001–02, ONSET, USA). The resolution should be 0.41 cm, not 0.21 cm.

Response: We have corrected the error in the revised manuscript.

4. p.13 l.210: The wire length of the temperature sensor (HOBO S–TMB–M0017, ONSET, USA) is only 17 m. Just out of curiosity: how to measure the temperature to a depth of 30 m?

Response: Extension cables (HOBO S–EXT–M025, ONSET, USA) were used to measure temperatures at depths greater than 17 m. We have added the description for the use of extension cables in the revised manuscript.

5. p.13 l.213: Please check the resolution of the temperature sensor (HOBO S–TMB–M0017, ONSET, USA). The resolution should be $\pm0.03$ ℃, not 0.03 ℃.

Response: Change was made as suggested.

6. p.14 l.224 and p.15 l.233: It is tough to see the differences described in your text from Figures 3 and 4 because the x-axis covers a range as long as six years. Please consider adding another plot or subplot with a shorter time span.

Response: We have added the subplots showing the change of groundwater table depth and temperature within one hydrological year in Figure 6 and 7 in the revised version.

[Figure]

Figure 6. The left panels show the dynamics of groundwater depth over six consecutive years, while the right panels highlight the variation in groundwater depth within a hydrological year.

[Figure]

Figure 7. The left panels show the dynamics of ground temperature at different depths over six consecutive years, while the right panels highlight the variation in ground temperature at different depths within a hydrological year.

7. p.16 l.260: Please double-check the pore diameter of the filter membrane used for DOC water sampling. It should be 0.7 μm.

Response: We have further double-checked the field records for sampling and confirmed that the 0.45 μm pore-diameter of the filter membrane was used. This size pore diameter of filter membrane is generally used to differentiate DOC from particulate organic matter (POC) (Wangersky, 1993; Zsolnay, 2003).

References

Wangersky, P. J.: Dissolved organic carbon methods: a critical review, Mar. Chem., 41, 61-74, https://doi.org/10.1016/0304-4203(93)90106-X, 1993

Zsolnay, Á.: Dissolved organic matter: artefacts, definitions, and functions, Geoderma, 113, 187-209, https://doi.org/10.1016/S0016-7061(02)00361-0, 2003.

8. p.18 l.291: 'ppb' is generally not used to represent the unit of measurement precision. It is suggested to be replaced by 'ug/L'.

Response: Change was made as suggested.

9. p.18 l.300: A space is missing in 'of' and '$^{13}$C'.

Response: Change was made as suggested.

10. p.28 l.590 and p.28 l.608: The URLs of the two references cannot be opened. Please double-check.

Response: After inspection, only the URL of the reference in p.28 l.608 cannot be opened. Therefore, we have changed the URL that can be opened in the revised manuscript, as follows: http://en.cnki.com.cn/Article_en/CJFDTOTAL-ZRZZ201303004.htm.

**The dataset**

These comments concern only the dataset itself given at https://doi.org/10.5281/zenodo.5184470

1. There seems to be a language problem with the title of the dataset. Please correct it.

Response: We have corrected the error. Please refer to the latest version of the dataset (https://doi.org/10.5281/zenodo.6296057).

2. Please add elevation data for each site in the datasets.

Response: We have added elevation data for each site. Please refer to the latest version of the dataset (https://doi.org/10.5281/zenodo.6296057).

3. Please simplify the analysis results. For example, the DOC concentration in precipitation was not measured using the total organic carbon analyzer (Multi N/C 2100 TOC, Analytik Jena AG, Germany). Thus, this column can be deleted from the datasets.

Response: We have simplified the dataset and deleted the column of the DOC concentration in precipitation. Please refer to the latest version of the dataset (https://doi.org/10.5281/zenodo.6296057).

4. It seems that some groundwater level data were missing. Please explain the reasons for the data missing

(please refer to Minor comment #4).

Response: Thanks for your suggestion. Some missing groundwater level data is because that the groundwater tables were deeper than wells screen depths during cold periods and thus not shown in the figure.

**Response to Reviewer 2:**

**Major issues:**

1. Although the ESSD aims to provide high quality datasets for the earth science field, it is still a scientific journal to present the new results, knowledge and understanding based on obtained dataset, rather than simply describing the measurements of datasets. The manuscript in this stage seems like a field experiment report, but lack the critical analysis for datasets, the knowledge indicated from datasets and understanding implicated from datasets.

For example, the groundwater level variations from four well groups indicate that deeper depth has the larger variations (over 15 m) of groundwater head (Figure 3), which implicates the mountainous groundwater level particular in high elevation would be very sensitive to changes in land surface or hydrogeology structure (thawing permafrost will reshape the hydraulic connectivity). All of these observations are worthy of in-depth analysis and discussion.

Therefore, I would like to suggest restructuring the section 3 and 4, presenting the data with a clear result analysis and discussion part, to demonstrate how these data help improving understanding for surface-groundwater interactions of Tibet Plateau, and what insights to provide for other alpine regions.

Response: Thanks for your suggestion. Indeed, all of the observation data are worthy of in-depth analysis and discussion since we present the systematic information for groundwater level, ground temperature, precipitation, river discharge, chemical and isotopic compositions of different water bodies at the catchment scale. These data are critical for understanding the groundwater flow and its interaction with surface water under impact of freeze-thaw process, and also other hydrological cycle-related issues such as biogeochemical processes. However, the aim & scope stated by the ESSD journal is "Earth System Science Data (ESSD) is an international, interdisciplinary journal for the publication of articles on original research data (sets), furthering the reuse of high-quality data of benefit to Earth system sciences. …**Any interpretation of data is outside the scope of regular articles**" (https://www.earth-system-science-data.net/about/aims_and_scope.html). Thus, the main purpose of this manuscript is to provide the high-quality datasets for understanding the surface-groundwater interactions on the TP and for exploring the different kinds of groundwater flow-related issues. If we add discussion part, we would inevitably have to interpret the data, which is outside the scope of regular articles in the ESSD. However, we still revised our manuscript to clearly present the monitoring system design, the method for data collection, water sample preservation analysis etc. We also present the results for spatiotemporal change of groundwater level, ground temperature, and river discharge, chemical and isotopic compositions in the catchments. The other results for the lithology of the aquifer and the mineralogical compositions were also included.

2. The authors give detailed descriptions for observed process, but lack a clear scientific objective for experimental design. Readers will be wondering, such as, why they put wells in those four locations, why they collected samples from western tributary…

Thus, a clear objective corresponding to measurement should be need, which will strongly demonstrate the significance of the study and datasets.

Response: The general principle behind the design of the field monitoring system is that the system can be used to obtain different kinds of hydrological and hydrogeochemical data for various hydrological components within the catchment such as glacier-snow meltwater, groundwater, and river water during different periods of the freeze-thaw cycle and at different locations along primary flow paths. With these data, it is possible to explore the hydrological processes and associated biogeochemical processes under the influence of the freeze-thaw cycle at the catchment scale.

The motivations for designing monitoring systems were added in the revised manuscript. One set of cluster well is located in the permafrost zone, so that the groundwater level, temperature, chemical and isotopic components of different waters in permafrost zone could be monitored. Due to the harsh and difficult field condition, the drilling instruments can't be transported to other sites in permafrost zone during our field study. Thus, only one set of wells in permafrost were drilled. The three sets of cluster wells in seasonal frost zone were designed to locate in the recharge, flow through and discharge zone of the sloping piedmont plain. With this design, it is possible to understand the aquifer system in both permafrost and seasonal frost zone, connection of flow and associated solute transport from permafrost zone to seasonal frost zone, and further to catchment outlet.

**Specific issues:**

1. Title: "research on groundwater flow and its interactions with surface water" is so general that easy to lose readers who hope to find relevant datasets. Moreover, the manuscript lacks the in-depth analysis of how these data indicate the groundwater flow and surface-groundwater interaction. So I would like to suggest a direct and clear title, such as "Dataset for alpine groundwater levels and hydrogeochemistry in northern Tibetan Plateau".

Response: Please refer to the response to the Major issues #1. Interpretation of the dataset to understand the groundwater flow and surface-groundwater interaction is out of the journal's scope. We revised the title to "Multi-year dataset for groundwater level, temperature, and chemical and isotopic compositions of different water bodies in an alpine catchment on the northeastern Qinghai-Tibet Plateau, China" in the revised manuscript.

2. Lines 58-59: Actually there are some groundwater studies focused on TP, and need to be included.
*Yao, Y., et al. (2017). "What Controls the Partitioning between Baseflow and Mountain Block Recharge in the Qinghai-Tibet Plateau?" Geophysical Research Letters 44(16): 8352-8358.*
*Yao, Y., et al. (2021). "Role of Groundwater in Sustaining Northern Himalayan Rivers." Geophysical Research Letters 48(10).*
Response: We have cited the above paper in the revised manuscript.

3. Line 94: "Groundwater level and ground temperature changes are also explained". Explain what?

Maybe it is better to use "show".

Response: Change was made as suggested in the revised manuscript.

4. Line 100: Since the whole manuscript did not mentioned the significance of this study for the Heihe River, I would like to suggest highlighting that this is a typical case for permafrost regions or headwater areas of TP.

Response: The selected catchment (Hulugou catchment) as one of the typical headwater catchments of Heihe River Basin on the TP, is addressed in the followings. Some of the following information was given in the appropriate locations of the manuscript.

(1) The Hulugou catchment, located in the headwater region of the Heihe River (HR), has a drainage area of 23.1 km$^2$. The HR headwater region is composed by many small catchments including Hulugou catchment. We have examined the catchments in which the water directly discharge to the Heihe River in the headwater region based on field investigation and remote sensing data, and 34 catchments have been identified with an total area of 1978.5 km$^2$. Among them, 28 catchments have an area within a range from 10 to 100 km$^2$, which means Hulugou catchment has a comparable size with most of the catchments. This would be helpful for quantifying the annul flow discharging from these catchments.

(2) The Hulugou catchment is representative of a series of topography features, including glacier, permafrost, and seasonally frozen area (Chen et al., 2014). Groundwater and surface water is directly recharged by the glacier-snow meltwater, and the aquifer is a complex condition affected by both permafrost area and seasonally frozen area. These phenomena are common in the catchments of the HR (Chen et al., 2006; Cheng and Jin, 2013; Cuo et al., 2014).

(3) The Hulugou catchment is a topographically closed catchment, with bedrock outcropping in the south mountains, and thick and widely spread Quaternary loose sediments depositing in the north piedmont alluvial plain. This kind of "mountain-piedmont plain" catchment is common in the headwater region of the Heihe River. The moraine deposits-talus-alluvial-pluvial deposits complexes form the main groundwater aquifers in the HR basin and play an important role in regulating groundwater discharge.

References
Chen, R., Song, Y., Kang, E., Han, C., Liu, J., Yang, Y., Qing, W., and Liu, Z.: A cryosphere-hydrology observation system in a small alpine watershed in the Qilian Mountains of China and its meteorological gradient, Arct. Antarct. Alp. Res., 46, 505-523, https://doi.org/10.1657/1938-4246-46.2.505, 2014.
Chen, Z., Nie, Z., Zhang, G., Wan, L., and Shen, J.: Environmental isotopic study on the recharge and residence time of groundwater in the Heihe River Basin, northwestern China, Hydrogeol. J., 14, 1635-1651, https://doi.org/10.1007/s10040-006-0075-7, 2006.
Cheng, G. and Jin, H.: Permafrost and groundwater on the Qinghai-Tibet Plateau and in northeast China, Hydrogeol. J., 21, 5-23, https://doi.org/10.1007/s10040-012-0927-2, 2013.
Cuo, L., Zhang, Y., Zhu, F., and Liang, L.. Characteristics and changes of streamflow on the Tibetan Plateau: A review. J. Hydrol.-Reg. Stud., 2, 49-68. https://doi.org/10.1016/j.ejrh.2014.08.004, 2014

5. Line 104: should be "annual averaged precipitation"
Response: Change was made as suggested in the revised manuscript.

6. Line 107: Since the daily discharge is much highly varied, I would like to convert the volume unit of (m³/day) to depth (mm/year).

Response: According to the comment, we have converted the volume units of (m³/day) to runoff depth (mm/year).

The sentence read now as:

"*The discharge from the catchment was approximately 567.7 mm/year in 2012 (Chen et al., 2014a; Chen et al., 2014b).*"

7. Line 121: Should be "good hydraulic connectivity"

Response: Change made as suggested.

8. Line 124: Since the precipitation involves the snow, this should be revised as "the aquifer is recharged by rainfall, melt water from glaciers and snow".

Response: Change was made as suggested in the revised manuscript.

9. Line 126: How low of the vegetation coverage? This should provide a percentage value at least. And this sentence is inconsistent with the following descriptions on vegetation coverage in line 135 (shrubs) and 147 (meadows). All these three parts should be combined and presented in consistency and with a clear percentage.

Response: We have combined these three parts and consistently presented the vegetation coverage with a percentage value in the revised manuscript.

The sentences read now as:

"*The permafrost zone (3500–4200 m a.s.l.) is dominated by alpine meadow, and the vegetation coverage is ~80 % (Chen et al., 2014b; Yang et al., 2015). The seasonal frost zone (below 3500 m a.s.l.) is dominated by alpine meadow and alpine shrubs, and the vegetation coverage is ~95 % (Liu et al., 2012; Chen et al., 2014b; Yang et al., 2015).*"

References

Chen, R., Song, Y., Kang, E., Han, C., Liu, J., Yang, Y., Qing, W., and Liu, Z.: A cryosphere-hydrology observation system in a small alpine watershed in the Qilian Mountains of China and its meteorological gradient, Arct. Antarct. Alp. Res., 46, 505-523, https://doi.org/10.1657/1938-4246-46.2.505, 2014.

Liu, Z., Chen, R., Song, Y., and Han, C.: Characteristics of rainfall interception for four typical shrubs in Qilian Mountain, Acta Ecologica Sinica, 32, 1337-1346, https://doi.org/10.5846/stxb201012211822, 2012.

Yang, F., Huang, L., Li, D., Yang, F., Yang, R., Zhao, Y., Yang, J., and Liu, F.: Vertical distribution of soil organic and inorganic carbon and their controls along topsequences in an alpine region, Acta Pedologica Sinica, 52, 1226-1236, https://doi.org/10.11766/trxb201504220193, 2015.

10. Line 128-129: how about the active layer thickness, 2 m?

Response: The active layer is ~2 m thick. We have clarified this sentence in the revised manuscript.

11. Line 140: Is this unconfined aquifer?

Response: Yes, this is an unconfined aquifer.

12. Figure 1: I would like to suggest adding one or two photos to show your field experiments.

Response: Thanks for your suggestion. We have added the pictures for each typical landform and the well layout as shown below (Fig. 1 and Fig. 2) in the revised manuscript.

[Figure]

Figure 1. Pictures showing (a) the glaciers in the south part of the study area, (b) the moraine sediments in the periglacial zone, (c) the planation surface in the permafrost zone, and (d) the piedmont alluvial plain in the seasonal frost zone.

[Figure]

Figure 2. Pictures showing the field scenes for (a) the location and layout of the four well groups in the study area, (b) the layout of wells in the well group WW01, (c) the layout of wells in the well group WW02, (d) the layout of wells in the well group WW03, and (e) the layout of wells in the well group WW04.

13. Line 155: "Well groups" will make readers misunderstanding there are multiple wells in a group. Direct using well would be better, and indicate one well includes multiple boreholes in different depth.

Response: Here we did mean that one well group includes multiple depth-specific wells in a group instead of multiple boreholes in different depth.

14. Line 181: Is this clay layer the top of the confined aquifers?

Response: Since we only have four sets of cluster wells, we are not sure if the clay layer is continuously distributed throughout the aquifer and can't confirm if this clay layer is the top of the confined aquifer. But our groundwater level data don't indicate that the porous media aquifer is confined aquifer.

15. Line 218-220: Should give a detailed discussion for WW04, because it is the only well in the permafrost area.

Response: We have added a detailed description of well sets WW04 in the new **Section 3** of the revised version. Please refer to the response to the Major issues #1.

16. Figure 3: The caption should note this is "daily variations"

Response: Change was made as suggested in the revised manuscript.

17. Section 4.2 and 4.3: Any results and information we obtained from these part of datasets?

Response: We have added results and information obtained from these part of datasets in the new **Section 5** of the revised version. Please refer to the response to the Major issues #1.

---

## Author Response (AR2)

Dear Editor,

We are submitting our revised manuscript essd-2021-273 "Integrated hydrogeological and hydrogeochemical dataset of an alpine catchment in the northern Qinghai-Tibet Plateau" to your prestigious journal.

We would like to thank you for your valuable comments, which have improved the presentation and quality of this work. We have revised our manuscript according to your comments. A point-by-point response to each of the comments is also provided in the reply letter.

We hope that you find this revised manuscript is suitable for publication in your journal.

Thank you very much for handling our manuscript.

Sincerely,

Rui Ma on behalf of all authors
* * *
**Response to Comments:**

1) The title seems too long and consists of too much information. A complex title could also confuse the audience. I would suggest the authors to consider shortening it to highlight only the key summary of the dataset.

Response: We shortened the previous title to " Integrated hydrogeological and hydrogeochemical dataset of an alpine catchment in the northern Qinghai-Tibet Plateau" in the revised manuscript.

2) The letter "S" was labeled in many of the figures. I suppose it was pointing the south direction. Please clarify it in the captions.

Response: To address this comment, we have clarified the letter "S" in the captions of the figures, as shown in Figure 1 and Figure 2.

[Figure]

Figure 1. Pictures showing (a) the glaciers in the south part of the study area, (b) the moraine sediments in the periglacial zone, (c) the planation surface in the permafrost zone, and (d) the piedmont alluvial plain in the seasonal frost zone. The letter "S" denotes the south direction.

[Figure]

Figure 2. Pictures showing the field scenes for (a) the location and layout of the four well groups in the study area, (b) the layout of wells in the well group WW01, (c) the layout of wells in the well group WW02, (d) the layout of wells in the well group WW03, and (e) the layout of wells in the well group WW04. The letter "S" denotes the south direction.

3) If it is possible, I would encourage the authors to disclose the location and date of the photos shown in the figures.

Response: We have added the location and date of photos, as shown in Figure 1, Figure 2, and Figure 3 below.

[Figure]

Figure 3. Pictures showing procedures for sampling (a) river water , (b) glacier meltwater, (c) spring water, and (d) groundwater in the field. Devices for collecting snow meltwater and precipitation were shown in (e) and (f), respectively.